



# Evaluation of nitrogen oxides sources and sinks and ozone production in Colombia and surrounding areas

Johannes G.M. Barten[1], Laurens N. Ganzeveld[1], Auke J. Visser[1], Rodrigo Jiménez[2], and Maarten C. Krol[1,3]

[1]Wageningen University, Meteorology and Air Quality Section, Wageningen, the Netherlands
[2]Department of Chemical and Environmental Engineering, Air Quality Research Group, Universidad Nacional de Colombia – Bogotá, Colombia
[3]Institute for Marine and Atmospheric Research Utrecht, Utrecht University, Utrecht, the Netherlands

**Correspondence:** Johannes G.M. Barten (sjoerd.barten@wur.nl)

**Abstract.** In Colombia, industrialization and a shift towards intensified agriculture have led to increased emissions of air pollutants. However, the baseline state of air quality in Colombia is relatively unknown. In this study we aim to assess the baseline state of air quality in Colombia with a focus on the spatial and temporal variability in emissions and atmospheric burden of nitrogen oxides ($NO_x = NO + NO_2$) and evaluate surface $NO_x$, ozone ($O_3$) and carbon monoxide ($CO$) mixing ratios. We

quantify the magnitude and spatial distribution of the four major $NO_x$ sources (lightning, anthropogenic activities, soil biogenic emissions and biomass burning), by integrating global $NO_x$ emission inventories into the mesoscale meteorology and atmospheric chemistry model WRF-Chem. The comparison with in situ measurements is bound to urban areas whereas the use of remote sensing data allows to also evaluate air quality in remote regions. WRF-Chem was set up for a domain centered over Colombia with a similar resolution as OMI observed $NO_2$ vertical columns as well as the EDGAR anthropogenic

emission inventory, both providing information on a ∼20 km resolution. However, this apparently poses a challenge regarding comparison with these urban observations. Air mass factors were recalculated based on the vertical distribution of $NO_2$ within WRF-Chem, with respect to the coarse ($1°\times1°$) a priori profiles because WRF-Chem is expected to better resolve spatial contrasts in $NO_2$ profiles. The main reason for recalculation is a more consistent satellite-model comparison but it also reduced the mean bias. WRF-Chem was, on average, able to provide good estimates for tropospheric $NO_2$ columns with an

averaged difference of $0.02\cdot10^{15}$ molecules $cm^{-2}$, which is <5% of the mean column. However, the simulated $NO_2$ columns are overestimated in regions with abundant modeled lightning emissions and underestimated in regions where biomass burning emissions dominate in the model. This result reflects the high contribution by lightning emissions (1258 Gg N $yr^{-1}$), even after already significantly reducing the emissions, and the low contribution by biomass burning emissions (104 Gg N $yr^{-1}$) to total $NO_x$ emissions within the WRF-Chem domain. WRF-Chem was unable to capture $NO_x$ and $CO$ urban pollutant mixing ratios,

both in timing and magnitude. Yet, WRF-Chem was able to simulate the urban diurnal cycle of $O_3$ satisfactory but with a systematic overestimation of 10 ppb due to the equally large underestimation of $NO$ mixing ratios and, consequently, titration. This indicates that these city environments are in the $NO_x$-saturated regime with frequent $O_3$ titration. We also applied an online meteorology-chemistry single column model (SCM) to evaluate how enhanced emissions and different representation of advection and mixing conditions could explain an improved representation of the observed $O_3$ and $NO_x$ diurnal cycles. The





25 SCM appears to indeed better represent the observed diurnal cycle of urban pollutant mixing ratios. But, interestingly, this result did not require an enhancement in the emissions, indicating that the role of boundary layer dynamics and advection should be considered besides the use of high-resolution models and emissions inventories to realistically simulate urban air quality. Overall, the presented approach shows a concise method, integrating in situ and remote sensing observations, to quantify air quality in regions with a limited measurement network. This study not only identifies four distinctly different source regions,

30 but also shows the interannual variability of these sources during the last one and a half decade. Furthermore, this study shows that with a critical consideration of advection and (nocturnal) boundary layer mixing, relatively coarse anthropogenic emission inventories can give reasonable results regarding the diurnal cycle of urban pollutant mixing ratios. It serves as a base to assess scenarios of future air quality in Colombia, or similar regions with distinct contrasting emission regimes, complex terrain and a limited air quality monitoring network, as a function of further industrialization and land use changes.

## 35 1 Introduction

Nitrogen oxides ($NO_x$ = NO + $NO_2$) are one of the main precursors of lower atmospheric ozone ($O_3$). Exposure to $NO_x$ has an adverse effect on human health on acute and long-term basis (Panella et al., 2000; Wolfe and Patz, 2002). Likewise, $O_3$ is toxic to humans (WHO, 2003) and can also reduce agricultural yields (Ashmore and Marshall, 1998). Therefore, accurate monitoring and predictions of surface concentrations of these air pollutants are key. Especially in densely populated regions air

40 pollution has been a major concern and is expected to even have larger impacts in the future due to the continuous urbanization and increasing emissions from for example traffic.

Anthropogenic $NO_x$ is produced in combustion processes and is an indicator of industrial activity and transportation as well as other anthropogenic activities like biomass burning and agricultural activities. Anthropogenic sources add up to ∼70% (∼50% industrial activity/transportation, ∼20% biomass burning) of the total global annual $NO_x$ emissions (Lamarque et al., 2010).

45 In addition to anthropogenic sources, natural sources contribute to total nitrogen budgets. NO emissions from soils add up to ∼12-20% of the global $NO_x$ emissions on a yearly basis (Bradshaw et al., 2000; Ganzeveld et al., 2002a; Jaeglé et al., 2005; Vinken et al., 2014). Lightning emissions are estimated to attribute on average 10-18% to the global yearly $NO_x$ emissions (Pickering et al., 2016). In the tropics (35°N - 35°S), anthropogenic activities (7.81 Tg N yr$^{-1}$), biomass burning (8.28 Tg N yr$^{-1}$), soil emissions (5.44 Tg N yr$^{-1}$) and lightning discharges (6.33 Tg N yr$^{-1}$) all contribute an approximately equal fraction to

50 the total $NO_x$ emission budget (Bond et al., 2002). A modeling study in the tropics must therefore provide accurate estimates of all these source categories.

In Colombia, where economy is thriving after a period of civil war (Vargas et al., 2015), further industrialization and intensified agriculture have already resulted in- and are expected to further increase- $NO_x$ emissions (Ganzeveld et al., 2010). Previously, Grajales and Baquero-Bernal (2014) aimed to assess the air quality of Colombia with a relatively coarse (2.5°x2.0°) 3D global

55 model (GEOS-Chem), whereas other studies focused mostly on air pollution of other compounds in cities using local emission inventories (Zárate et al., 2007; Kumar et al., 2016; González et al., 2018). Currently, there is a lack of understanding of the baseline state of air quality in Colombia on regional scale. Following from this, an application of inventories of the different





sources of NO$_x$ (and other pollutants) and covering both Colombia and its surrounding, upwind, areas can give valuable information about the current state of air quality in Colombia. This is also essential to determine how air quality might change

in the future, e.g., due to further urbanization and land use changes such as the conversion to oil palm (Vargas et al., 2015). Up until now, Colombia does not have an air quality monitoring network covering the entire country. Current measurement sites are mainly located in or close to the major cities. The rural areas, which are now undergoing rapid land use changes, do not have air quality stations nearby. This makes air quality monitoring for the whole country a challenging task. The use of satellite data, to observe species like NO$_2$ and CH$_2$O, is a valuable tool to fill the gaps and evaluate air quality in remote regions

(Bailey et al., 2006; Kim et al., 2009; Webley et al., 2012). However, satellite retrievals in the tropics are often limited by the presence of clouds.

During the last decades, computational advances have increased the possibility to conduct more detailed meteorology and air quality studies (Bauer et al., 2015). The recognition of the effects of chemical composition of the atmosphere on meteorology have stimulated the development of online coupled meteorology/chemistry models (Baklanov et al., 2014). Nowadays, these

models can be run for a large range of temporal and spatial scales. Not only the models, but also global emission inventories have considerably improved in spatial resolution during the last decades (González et al., 2018). Even though they may not provide enough spatial detail and heterogeneity for local scale (< 1 km) studies, e.g. to compare with in situ observations, they have provided essential information regarding emissions for regional scale ($\sim$20 km) studies (Saide et al., 2012; Ghude et al., 2013). In this study, rather than using high resolution urban emission inventories (e.g. González et al., 2018), we will

demonstrate the importance of boundary layer mixing and advection in the comparison of simulated and observed in situ measurements.

The primary objective of this study is to assess the current baseline state of air quality in Colombia, diagnosed with a focus on NO$_x$, using global emission inventories in a regional atmospheric chemistry model resolving the atmospheric chemistry and meteorology at a resolution comparable to that of the emission inventories. Furthermore, we evaluate surface NO$_x$, O$_3$

and CO mixing ratios in urban regions. We are aware that air quality in Colombia concerns are generally not limited to smog photochemistry mainly involving O$_3$-NO$_x$-VOC chemistry. Actually high concentrations of particulate matter might pose the largest risk to public health in many Colombian urban areas (Kumar et al., 2016). However, in this study we focus on NO$_x$ as an insightful metric to assess the spatial and temporal patterns in air quality in this region given its role in O$_3$ photochemistry as well as the availability of remote sensing observations to be integrated with a bottom-up model analysis. In this study

we use the Weather Research and Forecasting model coupled with Chemistry (WRF-Chem) (Grell et al., 2005). The model outcomes will be compared to in situ measurements and satellite retrievals to address the performance of the model both at the surface and integrated over the troposphere. This evaluation of surface and total column —using a highly resolving coupled meteorology-air quality model including the identification of different NO$_x$ sources— seeks to fill the gaps between local scale (González et al., 2018; Zárate et al., 2007; Kumar et al., 2016) and larger scale studies (Grajales and Baquero-Bernal, 2014).

This study also includes an evaluation of the interannual variability of air pollution for the different source regions during the last one and a half decade. This analysis is not only useful to address the representativeness of the performed simulation and





to identify the baseline state of air quality in Colombia but also justifying potential use of the modeling system to assess future changes in air quality using future anthropogenic emission and land use change scenarios (e.g. Ganzeveld et al., 2010).

## 2 WRF-Chem & its emission inventories

### 2.1 Model: WRF-Chem


In this study we use the WRF-Chem (Grell et al., 2005) version 3.7.1. WRF is a non-hydrostatic mesoscale numerical weather prediction model used for operational and research purposes. Figure 1 shows the WRF-Chem domain including cities and regions that we refer to in this research.

The simulation was set up for one domain with a spatial resolution of 27 km centered at 4.89 °N, 71.07 °W. The entire domain
consists of 100 grid points in both the North-South and the East-West direction with 60 vertical levels —in a sigma coordinate system— up to 50 hPa. The simulation length is one month, with a spin-up time of 24h, covering the whole month of January 2014. Selection of this study period is motivated by the fact that January is the dry season in Colombia where loss of remote sensing data due to presence of clouds is minimized (see Sect. 3.1.1). In addition, in Appendix A we show how the selected study period can be deemed being representative for the baseline state of air quality in Colombia. A more detailed analysis on
this is presented in Sect. 6.

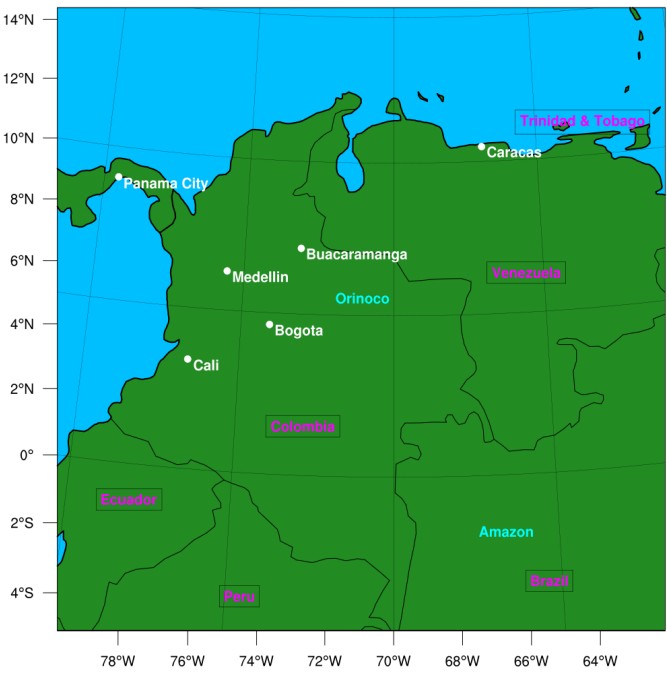

**Figure 1.** WRF-Chem domain including countries (pink), major cities (white) and regions (blue).





**Table 1.** WRF-Chem physical and chemical parametrization schemes.

| WRF-Chem option | Configuration |
| --- | --- |
| **Physical parameterizations** | |
| Microphysics | Morrison 2-moment (Morrison et al., 2009) |
| Long wave radiation | RRTM (Mlawer et al., 1997) |
| Short wave radiation | Dudhia (Dudhia, 1989) |
| Surface layer | Monin-Obukhov (Janić, 2001) |
| Land surface | Noah (Chen and Dudhia, 2001) |
| Boundary layer | YSU (Hong et al., 2006) |
| Cumulus | Grell 3D (Grell and Freitas, 2013) |
| Lightning option | P&R neutral buoyancy (Price and Rind, 1992) |
| **Chemical options** | |
| Gas-phase | CBM-Z (Gery et al., 1989; Zaveri and Peters, 1999) |
| Photolysis | F-TUV (Tie et al., 2003) |
| Lightning chemistry | Single-mode vertical distribution (Ott et al., 2010) |

The European Centre for Medium-range Weather Forecasting (ECMWF) provides us with the meteorological boundary conditions. The chemical boundary conditions are constrained with the Copernicus Atmosphere Monitoring Service (CAMS) near-real-time dataset. The boundary conditions are updated every six hours on a spatial resolution of 0.4° (∼44 km) with 60 vertical model levels. For January 2014, boundary conditions of $O_3$, $NO_x$, CO, $SO_2$ and $CH_2O$ are available. For tropospheric chemistry, the Carbon-Bond Mechanism version Z (CBM-Z) chemical scheme (Gery et al., 1989; Zaveri and Peters, 1999) is used here because it has been successfully implemented and tested in similar studies (Gupta and Mohan, 2015). Additional parametrization schemes used in this research are listed in Table 1.

## 2.2 Emission inventories

Anthropogenic emissions are described by the Emission Database for Global Atmospheric Research (EDGAR) dataset for greenhouse gases (Janssens-Maenhout et al., 2017) and Non-Methane Volatile Organic Compounds (NMVOCs) (Huang et al., 2017). Emission estimates are gridded on a 0.1°x0.1° resolution. EDGAR emissions are monthly estimates implying constant emissions over the whole simulation. In this study we use the EDGAR-HTAP emission inventory updated for 2010. EDGAR-HTAP uses nationally reported emissions combined with regional scientific inventories. For this research we assumed that 95% of the total anthropogenic emission of $NO_x$ is emitted as NO and 5% as $NO_2$ (Carslaw, 2005). VOC (Volatile Organic Compounds) speciation is according to Archer-Nicholls et al. (2014). In densely populated urban areas the anthropogenic emissions are dominated by vehicular emissions (Dodman, 2009). These emissions have a clear diurnal and weekly variation in contrast to emissions from the industry sector (Streets et al., 2003). Zárate et al. (2007) estimated traffic emission factors for Bogotá using in situ measurements and inverse modeling techniques. To account for this diurnal and weekly variation we





multiply the EDGAR emissions with the hourly and daily emission factors presented by Zárate et al. (2007).

The Global Fire Emissions Database version 4 (GFEDv4) dataset (Randerson et al., 2015) provides us with the biomass burning emissions. GFED is available on a spatial resolution of 0.25°x0.25°, approximately the same size as the WRF-Chem grid cells. Biomass burning $NO_x$ emissions are assumed to be completely in the form of NO.

Natural emissions of VOCs from terrestrial ecosystems are considered in this study using the Model of Emissions of Gases
and Aerosols from Nature version 2.1 (MEGANv2.1) (Guenther et al., 2012). Biogenic emissions are updated on-line using the WRF-Chem simulated surface temperature, soil moisture, leaf area index and photosyntetically active radiation. MEGAN also provides estimates of soil biogenic NO emissions.

The lightning-$NO_x$ parametrization scheme (Price and Rind, 1992), embedded in WRF-Chem, is used to account for $NO_x$ emissions by lightning. For this study we used an IC:CG (intracloud:cloud-to-ground) ratio of 2:1 constant over the whole
domain with a flashrate factor of 0.1. Per lightning flash (both for IC and CG strikes), it is assumed that 250 moles of NO are emitted (Miyazaki et al., 2014). It has to be noted that in an initial simulation, using standard WRF-Chem settings (flashrate factor = 1.0 & 500 moles of NO per strike), resulted in a significant overestimation of the lightning emissions (see Sect. 4.1) (Bradshaw et al., 2000; Miyazaki et al., 2014; Murray, 2016) and the settings we used resulted in a twentyfold decrease of lighting emissions compared to standard WRF-Chem settings.

# 3 Observations of atmospheric composition

## 3.1 Satellite retrievals

Observational data on the large scale distribution of $NO_2$ is retrieved from the Ozone Monitoring Instrument (OMI) onboard the National Aeronautics and Space Administration (NASA) Aura satellite (Levelt et al., 2006). OMI measures, among other pollutants, $NO_2$ column densities (Boersma et al., 2007) with daily, global coverage. The pixel size of 24x13 km$^2$ may be
coarse for particular applications, such as assessing urban pollution, but is suitable to assess contrasts in regional-scale air quality with apparent contrasting emission regimes. In addition, the resolution of the OMI observations is also comparable to the resolution of the anthropogenic emission inventory.

In this research we use the Quality Assurance for Essential Climate Variables (QA4ECV) $NO_2$ data product (Boersma et al., 2018). The measured slant columns —the tilted path directly from sun through the atmosphere to surface back to satellite—
are converted to vertical columns using Air Mass Factors (AMFs) [-] by

$$VCD = \frac{SCD}{AMF},\qquad(1)$$

where VCD and SCD are the Vertical Column Density and the Slant Column Density [molecules cm$^{-2}$], respectively. The AMFs define the relation between slant column and the vertical column above a pixel based on external information on e.g. surface albedo, scattering, clouds and the vertical distribution of $NO_2$ (Boersma et al., 2011). The vertical distributions of $NO_2$
in the QA4ECV product, which are used to calculate the AMFs, are simulated by the TM5-MP global chemistry transport model at a resolution of 1°x1° (Williams et al., 2017).





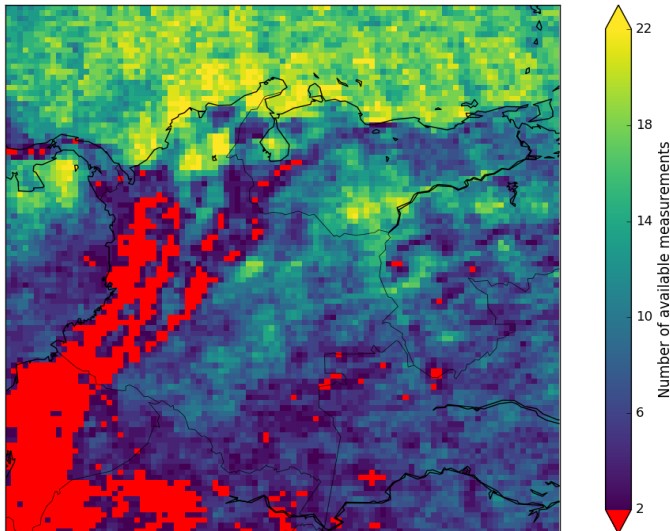

**Figure 2.** Spatial distribution of the available OMI measurements in January 2014 after filtering has been applied.

### 3.1.1 Filtering

We follow the data filtering recommendations by the QA4ECV consortium. Presence of clouds led to omission of 63% of OMI NO$_2$ data. Figure 2 shows the amount of OMI data per WRF-Chem grid cell after filtering the observations of January

2014. Especially above mountainous regions, where we also find the main urban areas of Bogotá and Medellín, there is a lack of available data due to the continuous presence of clouds limiting the quality of and which increases the uncertainty in the averaged tropospheric NO$_2$ column (Boersma et al., 2018). On average 9 data points per grid cell are available for this specific domain in January 2014, but with a large spatial heterogeneity. Some areas have >20 data points and other only two valid observations in this month.

### 3.1.2 AMF recalculation

The AMF dependens on assumptions of the state of the atmosphere and surface (e.g. surface albedo, cloud fraction, vertical distribution of NO$_2$) at the specific moment and location of a satellite observation (Lorente et al., 2017). This vertical sensitivity is described by an averaging kernel, which describes the relationship between the true column and the estimated, or retrieved column (Boersma et al., 2016). High-resolution models such as WRF-Chem are expected to better represent spatial gradients

in NO$_2$ profiles compared to coarse-scale global models such as GEOS-Chem or TM5-MP. Consqeuently, we can expect WRF-Chem to better resolve strong enhancements in tropospheric NO$_2$ VCDs in densely populated areas. Using grid sizes comparable to the size of such large urban areas is a major advantage of this procedure (Krotkov et al., 2017). The application of the averaging kernel is shown to reduce systematic representativeness errors for a satellite-model comparison (Boersma et al., 2016). We can recalculate the AMF based on the a priori concentration profile $x_a$ (from the TM5-MP model) and the



concentration profile in the high-resolution model $x_m$, in this study WRF-Chem (Boersma et al., 2016):

$$M'(x_m) = M(x_a) \frac{\sum_{l=1}^{L} A_l x_{m,l}}{\sum_{l=1}^{L} x_{m,l}}, \qquad (2)$$

where $M(x_a)$ is the tropospheric AMF used in the retrieval, $A_l$ are the elements of the averaging kernel for each $l^{th}$ vertical layer and $M'(x_m)$ is the recalculated AMF. In a next step, the new VCDs can be calculated by dividing the SCDs (retrieved by the satellite) with the recalculated AMFs (Eq. (1)).

Fig. 3 shows the difference in AMFs and the subsequent effect on the tropospheric $NO_2$ columns for the WRF-Chem domain. On average, we find a mean decrease in AMF of 0.05 with a standard deviation of 0.15. Regarding inferred changes in the VCD due to this recalculation of AMF, we find a mean increase in the VCD of $0.02 \cdot 10^{15}$ molecules cm$^{-2}$ ($\sim$3% of the average VCD) and a standard deviation of $0.07 \cdot 10^{15}$ molecules cm$^{-2}$. Above cities (e.g. Caracas, Bogotá, Medellín), we find mostly decreases in AMF (Fig. 3a). This indicates that there is more $NO_2$ present near the surface in WRF-Chem compared to TM5. This is

consistent with our expectation that WRF-Chem better captures the sub-1°x1° processes that are not resolved by TM5, such as the localized urban emissions. Furthermore, we find pronounced decreases in AMF above the Amazon region. However, these large decreases in AMF (up to -1) lead to an increase in the VCD of $0.5 \cdot 10^{15}$ molecules cm$^{-2}$ which is equal or even smaller than the increase in VCD due to the much smaller decrease in AMF over cities (Fig. 3b) because of the large VCDs over cities compared to over the Amazon region. Increases in AMF (Fig. 3a) are found mostly across the border from Colombia to

Venezuela, better known as the Orinoco region (Fig. 1). This reflects a higher abundance of $NO_2$ higher up in the troposphere from lightning sources. We also find two isolated hot spots of increases in AMF ($\sim$0.3) in southern Venezuela which correlate well with topography within the WRF-Chem domain which is less well resolved in the coarser resolution of TM5. Despite locally significant changes in VCDs, a domain average of $0.6 \cdot 10^{15}$ molecules cm$^{-2}$ indicates that the difference in the $NO_2$ a priori profiles of TM5 compared to those in WRF-Chem does not lead to domain-wide significant changes in VCDs.

### 3.1.3 Comparison of OMI and WRF-Chem

In this research, we focus on tropospheric $NO_2$ columns. In WRF-Chem we calculate the tropospheric $NO_2$ column by integrating from the surface to the tropopause, determined to be approximately the 50$^{th}$ model level ($\sim$90 hPa, $\sim$17km). This level is determined based on the average temperature profile (from surface to 50 hPa) of the complete simulation. Furthermore, to assess the daily differences in total $NO_2$ columns from OMI and WRF-Chem we need to co-sample their data points. For

Colombia, OMI passes around 17-19 UTC (1:00 PM local time). Grid points with none or only one measurement after filtering (see Fig. 2) will be completely discarded. In this way we aim to get a reliable comparison between WRF-Chem and OMI, which enables us to determine systematic biases in the regions dominated by different emission sources.





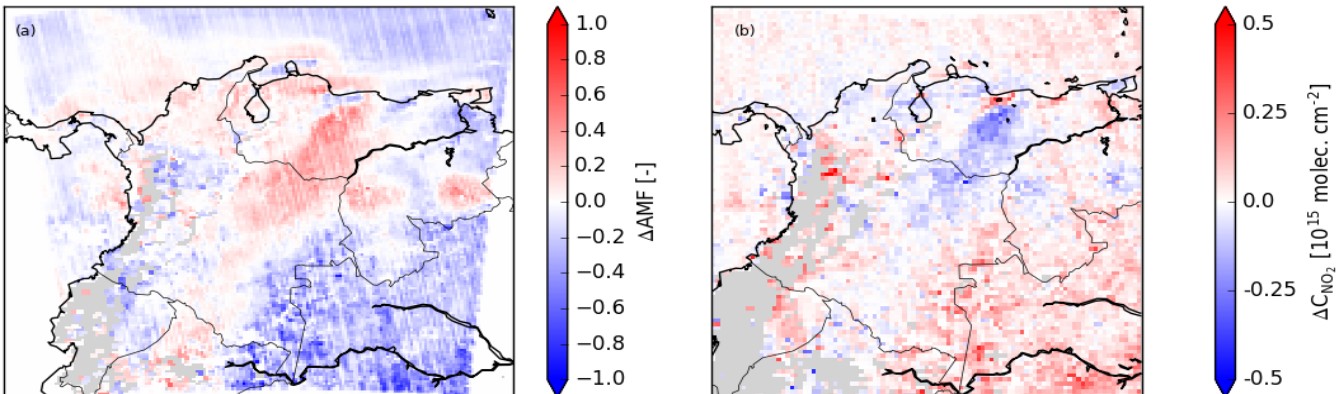

**Figure 3.** Spatial distribution of (a) the AMF difference (recalculated minus QA4ECV standard product, $\Delta$AMF [-]) in January 2014 based on the WRF-Chem simulation and (b) the subsequent effect on the NO$_2$ column difference (recalculated minus QA4ECV standard product, $\Delta C_{NO_2}$ [$10^{15}$ molecules cm$^{-2}$]) on the WRF-Chem grid.

## 3.2 In situ data

To further validate the model, besides the comparison with OMI observations, observational data from air quality monitoring
stations in Colombia are used. These include 30 stations confined to four cities in Colombia: Bogotá, Bucaramanga, Cali and Medellín (see Fig. 1). Observational data consists of 1-hourly averaged CO, NO, NO$_2$ and O$_3$ concentrations. The complete availability and locations of the station within the WRF-Chem domain can be found in Table B1. In this paper we only show the results of Bogotá also sincethe comparisons in other cities show similar results. Even on the still coarse resolution of the current WRF-Chem simulations, we expect that the evaluation of the temporal variability in simulated and observed concentrations
indicates how well the model captures some of the key drivers of atmospheric pollution.

## 4 Results

### 4.1 Nitrogen emission budgets and distribution

First of all, we identify the major sources of NO$_x$ within the domain of this study. The anthropogenic and biomass burning emissions are prescribed using their inventories whereas soil NO and lightning NO$_x$ emissions are explicitly simulated in
WRF-Chem. Some large cities contribute dominantly to the total NO$_x$ emissions (Fig. 4a). Total emissions are in the order of $\sim$10$^2$-10$^3$ Mg N month$^{-1}$ per grid box for the Colombian cities. However, largest NO$_x$ emissions, according to the EDGAR inventory, are found in and around Caracas, Venezuela. All these emissions can be attributed to anthropogenic emissions as reflected by a $\sim$100% contribution of anthropogenic emissions to the total emissions shown in Fig. 4b. Another major source of NO$_x$ is found in the south-east of the domain with values ranging up to 70 Mg N month$^{-1}$ per grid box. In this
region, dominated by rainforest, large convective systems are present generating thunderstorms with associated lightning NO$_x$





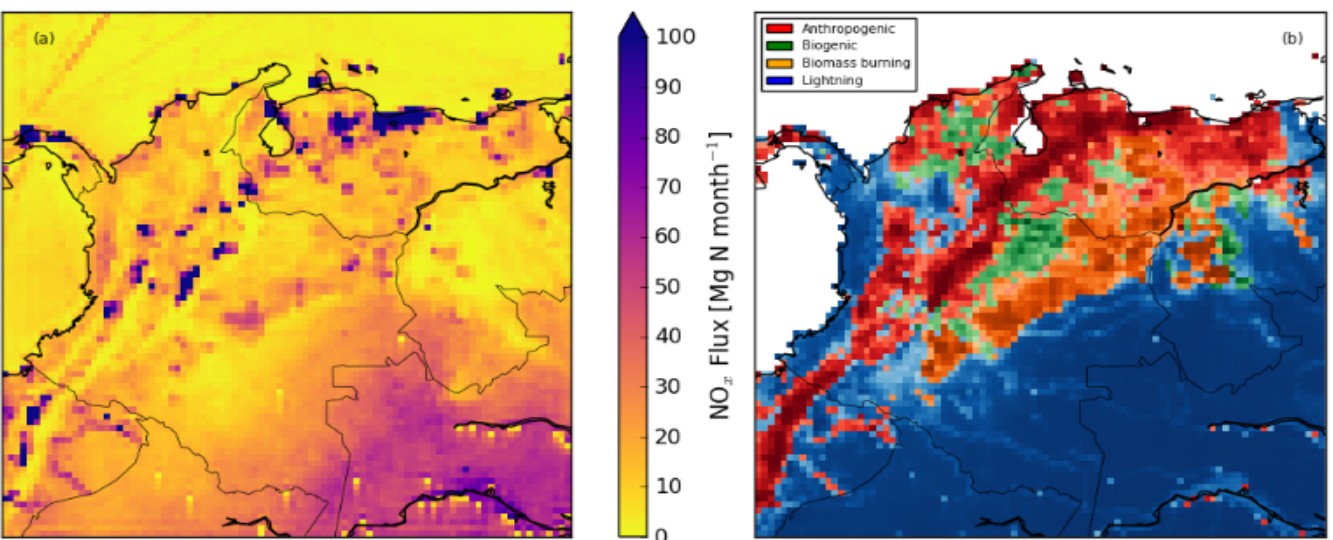

**Figure 4.** Spatial distribution of (a) the total NO$_x$ flux [Mg N month$^{-1}$] per grid cell in January 2014 and (b) the largest contributor [%] of the four emission inventories per grid cell over land. More saturated colours indicate a larger maximum fractional contribution up to 100%.

emissions. They appear to be the most important emissions in this region (Fig. 4b) also because anthropogenic and biomass burning emissions are mostly absent (with some exceptions near rivers). Biomass burning and soil biogenic emissions seem to be the most prominent sources of NO$_x$ across the Colombian-Venezuelan border (Fig. 4b), in the Orinoco region, in our model study. This region is dominated by savanna type grasslands which emit a relatively high amount of soil NO$_x$ but also

have a high probability of catching fire. NO$_x$ emissions in these regions are up to $\sim 10^1$-$10^2$ smaller compared to anthropogenic emissions, but, on the other hand, cover a larger area.

Lightning NO$_x$ emissions seem to be the most dominant emissions source over land in 63% of all grid cells the most dominant emission source, followed by anthropogenic- (22%), biomass burning- (9%) and biogenic (6%) emissions (Fig. 4b). Since we use four different emission inventories, all with their own estimates and uncertainties, the distinct contrasts in the spatial

distribution of emission sources will be key to determine spatially heterogeneous biases in satellite retrievals compared to WRF-Chem.

From budget calculations integrating over the whole domain, and using these January emissions to infer a NO$_x$ emission budget expressed per year (see Table 2), we find that lightning NO$_x$ emissions add up to 1258 Gg N yr$^{-1}$, with a distinct diurnal cycle. Anthropogenic NO$_x$ emissions add up to 933 Gg N yr$^{-1}$. Biogenic NO$_x$ emissions add up to 187 Gg N yr$^{-1}$ with daytime

emissions being $\sim$2 times larger compared to nighttime emissions (mostly regulated through temperature). Biomass burning NO$_x$ emissions —according to the emission inventory— provide the smallest contribution to the domain with a total NO$_x$ emission strength of 104 Gg N yr$^{-1}$.





**Table 2.** Total $NO_x$ emissions in the WRF-Chem domain per source category using January 2014 emissions to infer yearly total $NO_x$ emissions [Gg N yr$^{-1}$].

| $NO_x$ source category | Emission [Gg N yr$^{-1}$] |
|---|---|
| Lightning | 1258 |
| Anthropogenic | 933 |
| Biogenic | 187 |
| Biomass burning | 104 |
| Total | 2482 |

## 4.2  WRF-Chem & OMI comparison

To assess whether WRF-Chem is able to reproduce filtered and recalculated $NO_2$ VCDs satisfactorily we check for the spatial
and frequency distributions for both WRF-Chem and OMI (see Fig. 5). For WRF-Chem we find a wide range of column
densities (Fig. 5a). We find very low VCDs ($\sim 0.3 \cdot 10^{15}$ molecules cm$^{-2}$) over the Caribbean sea and across the eastern border
of Colombia into Venezuela. High VCDs in WRF-Chem are simulated above the major Colombian cities and the northeastern
part of the domain ($\sim 5 \cdot 10^{15}$ molecules cm$^{-2}$) while the highest VCDs are simulated above the city of Caracas with values up
to $8 \cdot 10^{15}$ molecules cm$^{-2}$. Similar to WRF-Chem, we find the lowest VCDs over the Caribbean sea in OMI (Fig. 5b). Also,
we find the highest VCDs above major cities —most pronounced for Caracas and Medellín— but the magnitude of the OMI
observed VCD ($\sim 2.5 \cdot 10^{15}$ molecules cm$^{-2}$) is much smaller compared to WRF-Chem. In OMI we find low VCDs above the
Amazon rainforest.

The large WRF-Chem VCDs ($\sim 5 \cdot 10^{15}$ molecules cm$^{-2}$) we find in the northeastern part of the domain (Fig. 5c) seem to reflect
mostly the role of the imposed boundary conditions which is not seen in the OMI retrievals where we only find a small plume
coming from Trinidad & Tobago transported westward due to the prevailing easterly wind. The overestimation above Caracas
might be due to an overestimation of anthropogenic emissions but this is not supported by a systematic major overestimation
above cities (for example, we find no overestimation above Panama City or Bucaramanga). However, the EDGAR emission
inventories are based on the year of 2010, when Venezuelan economy was still at its maximum (Wang and Li, 2016). After
2010, Venezuelan economy and oil production have declined strongly (Wang and Li, 2016) and therefore also emissions of
pollutants have been decreasing. In Sect. 6 we provide a more detailed overview of these findings regarding temporal changes
in Venezuelan emissions. Lastly, we find a systematic overestimation in the WRF-Chem simulated VCD above the Amazon
rainforest. Even though the overestimation is small in absolute terms ($\sim 0.5 \cdot 10^{15}$ molecules cm$^{-2}$) it is quite substantial relative
to the background mixing ratios. In this region, soil $NO_x$ release is small, anthropogenic activities are hardly present and
there are no known sources of biomass burning during January 2014. Consequently, overestimation in the simulated VCDs
can be attributed to the simulated major influence of lightning $NO_x$ emissions in this region (Fig. 4b). This further confirms
the finding that lightning $NO_x$ emissions are overestimated in WRF-Chem, even though they have already been significantly
reduced relative to the standard settings (see Sect. 2.2). However, we have to take into account that the OMI retrievals used for





this comparison reflect those conditions when cloud formation, and therefore lightning production, is less active resulting in very low VCDs. In contrast, the co-sampled WRF-Chem columns might reflect simulated cloud cover resulting in production

of NO by lightning. Nonetheless, the question whether lightning production was actually present or that it could not be picked up by OMI, being less sensitive to the presence of $NO_2$ below clouds, remains unanswered.

Remarkably, we find a region with systematic underestimations ranging from the center of Colombia to the northeastern border with Venezuela (the Orinoco region). In this region, there is no presence of major cities and lightning $NO_x$ emissions are small. The discrepancy we find might be due to missing agricultural- or biomass burning emissions. Localized enhancements

observed in OMI ($\sim2.5\cdot10^{15}$ molecules cm$^{-2}$) might also be caused by biomass burning emissions since enhanced soil $NO_x$ are expected to result in a more homogeneous enhancement of VCDs over a larger area with smaller intensities. We find that this intensity of biomass burning is not picked up by the WRF-Chem simulation using the GFED biomass burning inventory.

Figure 5d and Fig. 5e show, for both WRF-Chem and OMI, the scatter and frequency distribution in the $NO_2$ VCD. We find that both model simulated and observed VCDs show similar distributions, peaking at approximately the same VCD. However,

WRF-Chem shows more outliers especially regarding the simulation of high $NO_2$ VCDs. The 90% confidence interval of the WRF-Chem simulated VCDs is $(0.33,1.33)\cdot10^{15}$ molecules cm$^{-2}$ while for OMI the 90% confidence interval is $(0.32,1.06)\cdot10^{15}$ molecules cm$^{-2}$, with medians of $0.59\cdot10^{15}$ and $0.56\cdot10^{15}$ molecules cm$^{-2}$, respectively.

We find the median and mean of the absolute overestimation by WRF-Chem to be $0.02\cdot10^{15}$ and $0.09\cdot10^{15}$ molecules cm$^{-2}$ respectively (Fig. 5f). The 90% confidence interval equals $(-0.43,0.70)\cdot10^{15}$ molecules cm$^{-2}$. The distribution is approximately

Gaussian with a standard deviation of $0.53\cdot10^{15}$ molecules cm$^{-2}$ but somewhat left-skewed indicating an overestimation by WRF-Chem. This confirms the finding that WRF-Chem is able to produce on average good estimations for vertical $NO_2$ columns above Colombia. However, over- and underestimations can be significant, e.g. larger than the $\sim10\%$ uncertainty in monthly averaged OMI VCD over polluted regions (Boersma et al., 2018), due to numerous factors in both the model setup and the characteristics of the retrievals.

The use of the recalculated AMFs and VCDs, for a consistent model-satellite comparison (Sect. 3.1.2), also reduced the overall bias of the WRF-Chem simulated VCDs. We conducted the same analysis for the original OMI data. In this analysis we find that the 90% confidence interval for OMI VCDs changes to $(0.30,1.06)\cdot10^{15}$ molecules cm$^{-2}$ and the median and mean of the absolute error increase to $0.05\cdot10^{15}$ and $0.11\cdot10^{15}$ molecules cm$^{-2}$ respectively. Overall we get slightly larger columns (reflected by a increased 5[th] percentile) using recalculated columns and a reduction in absolute error between model and observations.

These results confirm the application of the recalculated OMI data.

### 4.3    Surface mixing ratios

We retrieve averaged diurnal cycles for the month of January shown in Fig. 6 by removing the significant spread in observed surface mixing ratios and by averaging the day-to-day variation in both the model and observations. We also compare the model simulated and observed temporal evolution over in $NO_x$ and $O_3$ over the whole simulation period (see Fig. C1, Appendix C).

WRF-Chem is able to represent the lowest (generally mid-day) $NO_x$ mixing ratios of individual stations quite well, but is unable to simulate the observed maxima (generally rush-hour) up to 200 ppb for particular events. Regarding $O_3$, WRF-Chem





**Figure 5.** Spatial distribution of averaged co-sampled Vertical Column Densities (VCD) [$10^{15}$ molecules cm$^{-2}$] for (a) WRF-Chem, (b) OMI recalculated retrievals and (c) the absolute difference between the two as well as (d) scatter plot, (e) frequency distribution of both WRF-Chem and OMI over the whole domain and (f) the distribution of the absolute difference between the two per grid point including mean (dashed line), median (dotted line) and 90% confidence interval (dashed-dotted line).





captures the upper limit of observed mixing ratios (∼40 ppb) but is unable to reproduce the low (<5 ppb) nighttime mixing ratios in the surface measurements.

Regarding simulated $NO_x$ we find an averaged diurnal cycle of 20 ppb during nighttime, with some day-to-day variation
(standard deviation = 10 ppb), and a minimum of 2 ppb during daytime but with less day-to-day variation (Fig. 6a). The observations reach peak mixing ratios around 7:00 local time where vehicular emissions during rush hour are mixed in a shallow boundary layer increasing $NO_x$ mixing ratios to 85 ppb on average. After rush hour mixing ratios decrease due to decreasing emissions, increasing boundary layer height and decreasing $NO_x$ lifetime. It is interesting to note that there does not seem to be a clear signal of evening rush hour in the $NO_x$ measurements and simulation.

The averaged diurnal cycle of CO in WRF-Chem shows a similar pattern to that of $NO_x$ (Fig. 6b). WRF-Chem shows daytime mixing ratios of ∼150 ppb (well above rural background mixing ratios of 100 ppb) and ∼350 ppb during nighttime while the surface measurements show a significantly larger variation. Averaged surface measurements during rush hour exceed CO mixing ratios of 1500 ppb indicating that the measurement stations are located above or near busy roads. Some measurement stations even report mixing ratios above 3000 ppb. Even though nighttime emissions are mixed over a smaller boundary layer
they appear to be considerably smaller so that surface mixing ratios remain lower. We find that WRF-Chem underestimates surface mixing ratios of CO by a factor of 4 during rush-hour and by a factor of 2 for nighttime conditions. These ratios are similar to the $NO_x$ ratios in Fig. 6a. Since CO has a relatively long lifetime compared to that of $NO_x$ we argue that observed differences regarding simulated and observed CO mixing ratios reflect issues regarding the representativity of the WRF-Chem grid simulated pollutant levels, including the representation of emissions and online simulated meteorological conditions,
relative to the footprint of the surface observations.

We find that for WRF-Chem most of the $NO_x$ is present as $NO_2$ with NO mixing ratios being very close to 0 ppb (Fig. 6c). In contrast, the observations show that most of the $NO_x$ is present as NO. For WRF-Chem we find a $[NO]/[NO_2]$ ratio of ∼0.32 (±0.13) during daytime and ∼0.07 (±0.04) during nighttime while for the surface measurements these ratios are ∼1.11 (±0.40) and ∼0.89 (±0.38) respectively. This dominance of NO in the $NO_x$ observations further indicates that the
measurement stations are situated very close to the main roads. The abundant fresh NO emissions at these locations quickly react with $O_3$ forming $NO_2$. The surplus NO, however, pushes the $[NO]/[NO_2]$ ratio up. Indeed, a simulated underestimation by WRF-Chem of 10 ppb NO during nighttime is consistent with a simulated overestimation of 10 ppb $O_3$ (Fig. 6d). We also find that in WRF-Chem, the formation of $O_3$ immediately starts at 6:00 local time (sunrise) while for the observations we find the lowest mixing ratios at 7:00 local time due to the extra NO titration caused by rush hour. Nonetheless, it seems that
chemical production and destruction rates of $O_3$, as well as other processes contributing to the overall magnitude and diurnal cycle in $O_3$, e.g., entrainment and deposition, are well captured by WRF-Chem considering the similar shape and amplitude of the diurnal cycle.



**Figure 6.** Averaged diurnal cycle of (a) NO$_x$, (b) CO, (c) NO and (d) O$_3$ mixing ratios [ppb] in Bogotá for the WRF-Chem output (black solid line) and averaged observational data (red solid line). The black and red shading indicate the 30-day standard deviation of WRF-Chem and observations respectively. The vertical lines, blue (night) and yellow (day) shading indicate daytime and nighttime.





## 5   Single Column Model

To test the hypothesis that the model-data mismatch over Bogatá is caused by a too coarse model resolution and representation
of emissions, we apply a Single Column Model (SCM). This SCM has been previously applied for an analysis of observations
of the plume of pollution downwind of the city of Manaus (Brasil) (Kuhn et al., 2010). In contrast to that study, conducting
so-called Lagrangian simulations with the SCM, we used here the SCM setup for a fixed location resembling the city of
Bogotá. The SCM explicitly considers atmospheric chemistry processes, including anthropogenic and natural emissions, gas-
phase chemistry, wet and dry deposition and turbulent and convective tracer transport as a function of meteorological and
hydrological drivers, surface cover, and land use properties (Ganzeveld et al., 2002b, 2008). For these urban area simulations
with the SCM we have modified the surface cover properties by prescribing surface roughness at 2 meters, assuming a small
vegetation fraction of 0.25, using a city area albedo of 0.1 and assuming reduced evapotranspiration (through a reduction of soil
moisture which limits transpiration). We nudged the SCM meteorology with wind speed, moisture, and temperature profiles
from the WRF-Chem simulation.
In order to simulate the chemistry in the SCM we constrained these simulations also nudging the concentrations of long-lived
tracers such as $O_3$, $NO_x$ and CO above the boundary layer using the CAMS data and used the same emissions, including
diurnal cycle, as in the WRF-Chem simulation. Using these settings in the SCM leads to a generally good agreement with the
observations in terms of 30-day average diurnal cycles, maximum early morning peak and daytime minimum mixing ratios
of CO, $NO_x$ and NO (Fig. 7). The SCM overestimates the rush-hour peak NO and $NO_x$ mixing ratios but is within the spread
of the observations. Especially during daytime, the simulated mixing ratios (~5-10 ppb NO and ~20 ppb $NO_x$) agree very
well with the observations. Regarding CO, the magnitude of the rush-hour peak is well represented (~1500 ppb). For both
the modeled and observed $NO_x$ mixing ratios we find maximum values at 7AM. For CO, the modeled maximum values are
also at 7AM, but the observations show a maximum at 8AM. The skewed $O_3$ diurnal cycle is also much better reproduced
compared to WRF-Chem although the maximum afternoon mixing ratios are equally overestimated. This SCM analysis shows
that the observations in Bogotá seem to be mostly governed by the interplay between emissions, boundary layer dynamics, and
chemistry, and that advection likely plays a limited role. With simulated substantially smaller wind speeds, the SCM simulates
almost every night the presence of an inversion. In contrast, the WRF-Chem simulations appear to also have quite efficient
mixing during the night due to simulated relatively high wind speeds preventing the build-up of a strong nocturnal inversion.
In the SCM, the simulated average nocturnal boundary layer height is in the order of ~90 meters while in the WRF-Chem
simulation this is ~200 meters. This additional model analysis indicates how in such direct comparisons of model simulated
and in situ urban area pollution levels, there should be a critical consideration of both the representation of spatial and temporal
variability in emissions as well as boundary layer dynamics and advection.

## 6   Discussion

The integration of global emission inventories in a highly resolved coupled meteorology-air quality model (WRF-Chem), with
roughly the same spatial scale, allowed us to assess the state of- and contribution by different sources to the air quality in Colom-



**Figure 7.** Averaged diurnal cycle of (a) NO$_x$, (b) CO, (c) NO and (d) O$_3$ mixing ratios [ppb] in Bogotá for the averaged observational data (red solid line) and the SCM runs (green solid line). The red and green shading indicate the 30-day standard deviation of observations and SCM respectively. The vertical lines, blue (night) and yellow (day) shading indicate daytime and nighttime.

... 

bia and neighbouring countries, diagnosed with focus on $NO_x$. We identified four major sources of $NO_x$ in Colombia which were implemented in WRF-Chem partly through emission inventories (anthropogenic and biomass burning) and partly through emission models (soil NO and lightning). Using January emissions to infer a $NO_x$ emission budget expressed per year we found that lightning $NO_x$ emissions are the main source for the domain applied in this study, with 1258 Gg N $yr^{-1}$. These are followed by respectively anthropogenic (933 Gg N $yr^{-1}$), soil biogenic (187 Gg N $yr^{-1}$) and biomass burning (104 Gg N $yr^{-1}$) emissions. Figure A1 shows the averaged VCDs over the regions dominated by one of the four emissions classes (Fig. 4b) to further evaluate how the presented combined WRF-Chem OMI $NO_2$ VCD analysis for January 2014 is representative for the $NO_x$ emissions for the larger study domain. The domain averaged anthropogenic or lightning dominated regions seem to have relatively low interannual variability. The biogenic and biomass burning dominated regions show most interannual variability which also seem to correlate with El Niño years (https://origin.cpc.ncep.noaa.gov/products/analysis_monitoring/ensostuff/ONI_v5.php, last access: 30 October 2019), with the exception of 2015. Colombia is relatively warm and dry during El Niño years (Córdoba-Machado et al., 2015). Figure A1 indicates that biogenic- and biomass burning emissions might have increased during El Niño years reflected by higher January monthly mean VCDs above those regions. Based on this further analysis of the long-term trends in OMI $NO_2$ VCDs, we can argue that the 2014 simulation is a reasonably good approximation of the baseline state of air quality in Colombia.

Using standard settings for the lightning $NO_x$ parametrization scheme (Ott et al., 2010) in WRF-Chem, emissions would be 20 times higher compared to the settings used in this study. In this research we reduced the predicted number of flashes tenfold and also reduced the number of moles NO emitted per lightning flash —which is still a major factor of uncertainty (Murray, 2016; Pickering et al., 2016)— from 500 to 250. Miyazaki et al. (2014) provides an overview of estimates of the amount of moles NO emitted per lightning flash based on satellite, laboratory, theoretical and field studies. Estimates range mostly from 10 to 650 moles NO per flash and rectify the use of 250 moles NO per flash in this study. Miyazaki et al. (2014) also estimated total lightning $NO_x$ emissions for a subdomain of South-America of 1.21 Tg N $yr^{-1}$, comparable to our results. However, these emissions are for a larger domain (38°x45°) covering a larger part of the Amazon rainforest compared to this study (23°x20°). In contrast, this study uses January emissions which are ∼50% larger than yearly averages for this region (Miyazaki et al., 2014), because of the dry season in Colombia generating more vigorous convection. The estimate of lightning produced $NO_x$ emissions for the small domain used in this study (1.26 Tg N $yr^{-1}$) already makes up a significant portion of the estimated worldwide lightning $NO_x$ emissions (2-12 Tg N $yr^{-1}$ (Murray, 2016; Bond et al., 2002; Bradshaw et al., 2000)) indicating that the lightning $NO_x$ parametrization scheme, despite the introduced significant decrease in flashes and amount of NO produced, still overestimates $NO_x$ emissions. Further attention is required regarding the lightning $NO_x$ parametrization scheme in follow-up studies on atmospheric $NO_x$ over Colombia, or other regions where lightning is a dominant source of $NO_x$.

Another noticeable outcome of this research is that biogenic and biomass burning emissions are ∼6 to 10 times smaller compared to lightning and anthropogenic emissions while other studies suggest that they would be of comparable magnitude in the tropics (Bond et al., 2002; Holland et al., 1999). Since all the emission inventories have been performing well in similar or larger scale studies (Ghude et al., 2013; González et al., 2018; Jiang et al., 2012; Zhong et al., 2016) we can not draw strong conclusions regarding a misrepresentation of the biogenic and biomass burning emissions as they also heavily depend



on the investigated domain. However, the spatial distribution and the relative importance of each emission inventory within the domain (Fig. 4b) provided us with valuable information for both the bottom-up validation using in situ data as well as the top-down validation approach using remote sensing data.

The top-down validation approach, using satellite retrievals, is a valuable tool to evaluate air quality in remote regions (Bailey et al., 2006; Webley et al., 2012) with a missing network of air quality monitoring in both urband and rural sites. The daily global coverage and retrievals of $NO_2$ by OMI (Levelt et al., 2006) were used to assess the quality of all emission inventories over the whole domain. However, 63% of the data is lost for this specific model setup mainly due to the continuous presence of clouds. Thus, longer simulation times have to be considered in the tropics compared to mid-latitudes. The vertical distribution of $NO_x$ within a modeling environment is key to identify discrepancies for a top-down validation approach using satellite retrievals. It has to be recognized that the satellite sensitivity is reduced towards the surface (Boersma et al., 2016), inducing enhanced differences between observed and modeled profiles. However, this can be overcome by replacing a priori TM5 profiles with those from the applied model (Boersma et al., 2016) that results in reduced the mean biases in this particular case as well.

WRF-Chem simulated VCDs showed the largest overestimation with respect to OMI above Caracas. This can be explained by a decrease of industrial activity in 2014 —with respect to 2010 EDGAR estimates— due to the decrease in economy and oil production in Venezuela (Wang and Li, 2016). This is further illustrated in Fig. 8 which shows the January monthly averaged VCDs for OMI from 2005 to 2019 over both the urban area of Bogotá and Caracas. We find an OMI observed January monthly averaged $NO_2$ column above Caracas of $2.1 \cdot 10^{15}$ molecules cm$^{-2}$ which agrees with the findings of Fig. 5b. However, the EDGAR anthropogenic emission inventory is based on the year 2010. For 2010 we find a January monthly averaged $NO_2$ column above Caracas of $3.4 \cdot 10^{15}$ molecules cm$^{-2}$ which explains part of the overestimation by WRF-Chem. This is supported by an estimated reduction in Venezuelan $CO_2$ emissions of ∼197 Tg $CO_2$ in 2010 to ∼183 Tg $CO_2$ in 2014 (http://edgar.jrc.ec.europa.eu/overview.php?v=CO2ts1990-2015, last access: 30 October 2019). The reduction in $CO_2$ emissions, as a proxy for $NO_x$ emissions, is mostly caused by a reduction in the industrial sector. From 2012 onwards, we find a clear declining trend in $NO_2$ columns caused by a decline in economic activity (Wang and Li, 2016). For Bogotá, the discrepancy between 2010 and 2014 columns is lower also indicated by a smaller bias found in Fig. 5 and indicate that the 2010 EDGAR emissions should reflect the baseline state of air pollution in Colombia reasonably well.

In contrast to the bottom-up validation approach, where WRF-Chem showed a significant underestimation of $NO_x$ compared to the in situ measurements, we found that WRF-Chem does not systematically underestimate urban VCDs. This suggests that the problem is indeed bound to representativeness of WRF-Chem with respect to sub-grid scale emissions and other processes and not so much to the magnitude of anthropogenic emissions. The underestimation by WRF-Chem in the Orinoco region, where biogenic and biomass burning emissions make up a great part of the emission budget, indicate an underestimation of biomass burning emissions. Biogenic emissions are expected to show a more homogeneous distribution over a larger area with less pronounced peak emissions although the role of enhanced emissions by pulsing and fertilizer application should not be ruled out (Ganzeveld et al., 2002a). Therefore, they are also not expected to explain VCDs over $2 \cdot 10^{15}$ molecules cm$^{-2}$ we found in OMI retrievals. This connects to the findings of Grajales and Baquero-Bernal (2014) who concluded that high VCDs in this





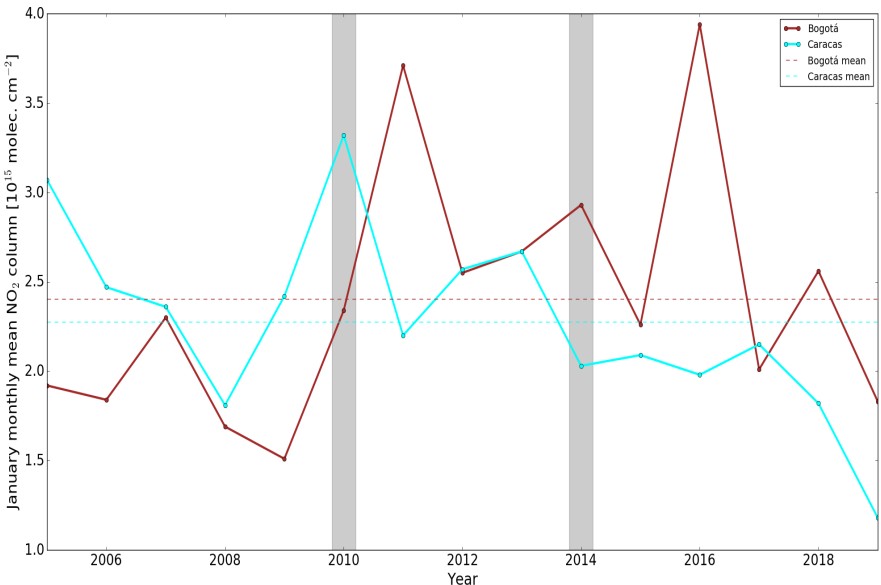

**Figure 8.** January monthly averaged $NO_2$ vertical column densities [$10^{15}$ molecules cm$^{-2}$] retrieved from OMI for 2005-2019 for the cities Bogotá (brown) and Caracas (cyan) including their mean (dashed) over the whole period. The grey vertical bars highlight the 2010 and 2014 years indicating the years of the EDGAR emissions and WRF-Chem simulation, respectively.

region are most likely related to biomass burning, which is apparently underestimated by the emission inventory we applied in this study. Castellanos et al. (2014) discussed that small fires could add up to 55% more burned area and that agricultural biomass burning $NO_x$ emissions may be significantly underestimated.

Connected to the budget calculations we find a large area with overestimations of VCDs in the region dominated by lightning

$NO_x$ emissions. These findings are in contrast with Grajales and Baquero-Bernal (2014) who found in their study with the GEOS-Chem modeling system that in remote regions without biomass burning there is an overestimation of OMI VCDs. Our study indicates that lightning $NO_x$ emissions are the major source of $NO_x$ that explains the discrepancy in the study by Grajales and Baquero-Bernal (2014) in which this source was not considered. Also, the use of WRF-Chem, having a spatial resolution approximately the same size as the OMI observations, can be advantageous over coarser models such as GEOS-Chem used by

Grajales and Baquero-Bernal (2014).

The air-quality monitoring network in Colombia is limited to four major cities. This implies that the validation is limited to urban areas where anthropogenic emissions are the dominant source of pollution. A comparison with in situ data showed that WRF-Chem systematically underestimates urban surface mixing ratios of $NO_x$ and CO. All the surface observations showed a clear signal of morning rush-hour emissions with average observed $NO_x$ mixing ratios up to 90 ppb and single observations not

rarely exceeding 150 ppb indicating that all surface monitoring stations are located at or near busy roads. We do not find any





evidence of evening rush-hour which is supported by Zárate et al. (2007) who estimated the temporal variability of vehicular emissions in Bogotá.

Similar to González et al. (2018), who focused on $O_3$ dynamics in Manizales (medium sized Andean city), we find an overestimation of $O_3$ by WRF-Chem both during nighttime and daytime. For Manizales, $NO_x$ measurements were not available

(González et al., 2018) and were proposed to explain most of inferred the discrepancies between the observed and simulated $O_3$ mixing ratios. In this study we found that the underestimation of NO by ∼10 ppb translates to an overestimation of ∼10 ppb $O_3$. Even though $O_3$ production and destruction is well captured by WRF-Chem, local emission inventories, including a more detailed spatial resolution around cities, can provide the extra detail needed for sub-grid scale analysis of the interactions between local-scale emissions, chemistry, mixing and resulting pollutant concentrations (González et al., 2018). But, as shown

in Sect. 5, a nested domain with local, high-resolution emission inventories is not always needed to resolve urban pollutant concentrations. With a different representation of advection and (nocturnal) mixing conditions, EDGAR emissions integrated in a relatively simple Single Column Model, can represent the averaged diurnal cycles of $O_3$, CO and $NO_x$ reasonably well.

One of the regions that is currently undergoing major land-use changes is the Orinoco. Its traditional agriculture and extensive grazing shift rapidly towards a more intensified production of food, biofuels and rubber (Lavelle et al., 2014). Especially oil

palm, which is one of the world's most rapidly expanding crops (Fitzherbert et al., 2008), is becoming more and more dominant in the Orinoco region (Vargas et al., 2015). Also, urbanization in Colombia is continuously increasing (Samad et al., 2012). Ongoing and anticipated future transformation of both rural and urban areas, in combination with expected increases in temperature and changes in the hydrological cycle, imply changes in emission budgets affecting air quality in the future. Further consistent coupling of land-use classes with emission representations, such as anthropogenic-, biomass burning-, biogenic-,

and lightning emissions apparently all having a generally dominant role in atmospheric $NO_x$ cycling in different regions of Colombia, may provide valuable information of future predicted air quality in Colombia.

## 7 Conclusions

This study presented an analysis of the baseline state of air quality in Colombia, focusing on $NO_x$ as main metric. Using a highly resolved coupled meteorology-air quality model (WRF-Chem), with roughly the same scale as both global emission

inventories as well as satellite retrievals (OMI), allowed us to identify sources of pollution and the baseline state of air quality in Colombia. The main findings illustrate that, within the modeling domain, lightning (1258 Gg N yr$^{-1}$), anthropogenic (933 Gg N yr$^{-1}$), soil biogenic (187 Gg N yr$^{-1}$) and biomass burning emissions (104 Gg N yr$^{-1}$) all contribute to the total nitrogen emission budget. Especially the spatial distribution, clearly identifying regions with different dominating $NO_x$ sources, shows the importance of providing good estimates of every individual source of $NO_x$ on its own.

The top-down validation approach, using OMI retrievals, showed that WRF-Chem was able to produce on average estimates of $NO_2$ Vertical Column Densities (VCDs) close to that observed. We found the mean and median of the difference between model and observations to be $0.02 \cdot 10^{15}$ and $0.09 \cdot 10^{15}$ molecules cm$^{-2}$, respectively. However, we found an overestimation of the lightning $NO_x$ production within WRF-Chem depicted by an overestimation of the vertical columns in the Amazon





region, where lightning $NO_x$ emissions are the only significant source of $NO_2$. Additionally, the comparison indicates that

biomass burning emissions are underestimated in WRF-Chem since OMI showed some strong enhancements in $NO_2$ not being reproduced by WRF-Chem. The biomass burning emission inventory shows some presence of wildfires in that region but the model only produces estimates of VCDs of $\sim1\cdot10^{15}$ molecules cm$^{-2}$, compared to OMI VCDs up to $2\cdot10^{15}$ molecules cm$^{-2}$, in regions where it is known to have significant biomass burning sources. Air Mass Factors (AMFs) were recalculated based on the vertical distribution of $NO_2$ within WRF-Chem with respect to the coarse ($1°x1°$) a priori profiles. The AMF recalculation

procedure, necessary to obtain a consistent comparison between WRF-Chem and OMI $NO_2$ columns, also resulted in a better agreement between model and satellite. Using recalculated AMFs decreased the median of the difference between WRF-Chem and OMI from $0.05\cdot10^{15}$ molecules cm$^{-2}$ to $0.02\cdot10^{15}$ molecules cm$^{-2}$ even though this was not the main reason for the recalculation. An analysis of the past one and a half decade of OMI $NO_2$ VCD data showed that the selected simulation period is representative for the baseline state of air quality in Colombia but also that the interannual variability in $NO_2$ columns over the

different source regions can be attributed to specific events such as ENSO.

The bottom-up validation approach using air quality monitoring stations in urban areas showed that WRF-Chem, at the relative coarse resolution, does not reproduce these observations given the role of large heterogeneity in the emissions and other processes determining pollution levels. Application of the anthropogenic EDGAR emission inventory ($0.1°x0.1°$ resolution) resulted in a clear underestimation of $NO_x$ and CO mixing ratios with respect to the local urban surface measurements. How-

ever, WRF-Chem was able to simulate the diurnal amplitude in $O_3$ reasonably well for all urban locations. It seems that the underestimation of $\sim10$ ppb $O_3$ both during day- and nighttime can be attributed to the underestimation of NO by $\sim10$ ppb. Application of a Single Column Model (SCM), to evaluate the impact of a modified representation of emissions based on the observed to WRF-Chem simulated CO mixing ratio, showed a much better agreement between observed and simulated surface mixing ratios. This was actually achieved using the EDGAR emissions as also applied in WRF-Chem and mainly due to es-

pecially a different representation of advection and (nocturnal) mixing conditions. This indicated that besides the use of local emissions inventories in highly resolved modeling systems, it is also essential to carefully assess the role of boundary layer dynamics, in partiuclar the representation of nocturnal mixing conditions, to evaluate simulations of pollutant concentrations.

In this study we presented a concise method, integrating both in situ and remote sensing observations with a mesoscale modeling system, to arrive at a quantification of air quality in regions with a limited measurement network to cover the large spatial

heterogeneity in air pollution source distribution. Results obtained in this study provide insight in the baseline state of air quality in Colombia. The findings add new information about uncertainties related to emission inventories and their application in regional air quality modeling. It may provide as a base for more local studies or the application towards future predictions of air quality in Colombia, due to land use changes, or comparable regions not having air quality monitoring networks with national coverage.

*Code and data availability.* OMI data, in situ data and WRF-Chem output are available upon request as well as scripts to recalculate the tropospheric AMF.





*Author contributions.* JGMB and LNG designed the experiment. JGMB performed the WRF-Chem simulations. LNG performed the single column model simulations. JGMB performed the analysis and wrote the manuscript, with contributions from all co-authors.

*Competing interests.* The authors declare no competing interests.

*Acknowledgements.* The authors acknowledge Oscar Julian Guerrero Molina from the Instituto de Hidrología, Meteorología y Estudios Ambientales (IDEAM) for providing us with air quality monitoring data. We acknowledge WRF-Chem developers and emission inventory (EDGAR, MEGAN, GFED) developpers. We also acknowledge the QA4ECV consortium for making OMI NO$_2$ data publicly available. We thank Folkert Boersma for his input on the OMI analysis.





## Appendix A: Long term analysis of OMI VCD

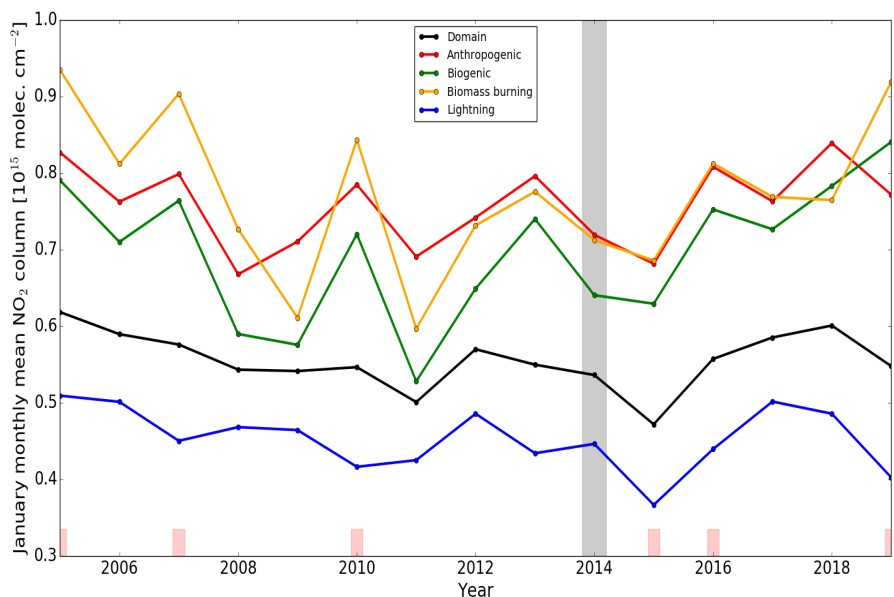

**Figure A1.** January monthly averaged $NO_2$ vertical column densities [$10^{15}$ molecules cm$^{-2}$] retrieved from OMI for 2005-2019 for the whole domain (black), regions with dominating anthropogenic (red), biogenic (green), biomass burning (yellow) and lightning (blue) emissions. The grey vertical bar highlights the WRF-Chem simulated year 2014. The red bars indicate El Niño years (2005, 2007, 2010, 2015, 2016, 2019).



## 520 Appendix B: Complete overview in situ data

**Table B1.** Available air quality monitoring stations including city, location and measured compounds.

| Station name | City | Latitude | Longitude | CO | NO | NO$_2$ | O$_3$ |
|---|---|---|---|---|---|---|---|
| Pance | Cali | 3.305 | -76.533 | | | | ✓ |
| Universidad del Valle | Cali | 3.378 | -76.534 | | | ✓ | ✓ |
| Compartir | Cali | 3.428 | -76.467 | | | | ✓ |
| C. Alto Rendimiento | Bogotá | 4.658 | -74.084 | ✓ | | | ✓ |
| Carvajal - Sevillana | Bogotá | 4.596 | -74.149 | ✓ | | | ✓ |
| Fontibon | Bogotá | 4.670 | -74.142 | ✓ | | | ✓ |
| Kennedy | Bogotá | 4.625 | -74.161 | ✓ | ✓ | ✓ | |
| Las Ferias | Bogotá | 4.691 | -74.083 | ✓ | | | ✓ |
| MinAmbiente | Bogotá | 4.626 | -74.067 | | | | ✓ |
| Puente Aranda | Bogotá | 4.632 | -74.118 | ✓ | ✓ | ✓ | ✓ |
| San Christobal | Bogotá | 4.573 | -74.084 | | | | ✓ |
| Tunal | Bogotá | 4.576 | -74.131 | ✓ | ✓ | ✓ | ✓ |
| Guaymaral | Bogotá | 4.784 | -74.044 | | ✓ | ✓ | ✓ |
| Suba | Bogotá | 4.761 | -74.094 | | ✓ | ✓ | ✓ |
| Usaquen | Bogotá | 4.710 | -74.030 | | | | ✓ |
| CAL-Corp. Lasallista | Medellín | 6.102 | -75.642 | | | | ✓ |
| ITA-Casa Justicia | Medellín | 6.188 | -75.601 | | ✓ | ✓ | |
| ITA-Col. Concejo | Medellín | 6.171 | -75.648 | | | | ✓ |
| MED-Politecnico JIC | Medellín | 6.212 | -75.581 | | ✓ | ✓ | |
| MED-Politecnico JIC (S) | Medellín | 6.212 | -75.581 | | ✓ | | |
| BEL-U.S. Buenaventura | Medellín | 6.331 | -75.569 | | ✓ | ✓ | ✓ |
| MED-Museo Antioquia | Medellín | 6.253 | -75.570 | ✓ | | | |
| MED-UN Fac. Minas | Medellín | 6.274 | -75.593 | | ✓ | ✓ | |
| MED-UN Nucleo Volador | Medellín | 6.266 | -75.580 | | ✓ | ✓ | ✓ |
| MED-Univ. Medellín | Medellín | 6.256 | -75.559 | | | | ✓ |
| MED-Villahermosa | Medellín | 6.256 | -75.559 | | | | ✓ |
| BAR-Parque Las Aguas | Medellín | 6.409 | -75.417 | | | | ✓ |
| Cabecera | Bucaramanga | 7.113 | -73.111 | ✓ | | | |
| Centro | Bucaramanga | 7.119 | -73.127 | ✓ | ✓ | ✓ | |
| Ciudadela | Bucaramanga | 7.106 | -73.124 | ✓ | | | |





## Appendix C:  O$_3$ and NO$_x$ mixing ratios in Bogotá for January 2014



**Figure C1.** Temporal evolution of (a) NO$_x$ and (c) O$_3$ mixing ratios [ppb] in Bogotá for WRF-Chem (black solid line) and all available observational stations (coloured points). Scatter plots of the WRF-Chem output compared with averaged (b) NO$_x$, (d) O$_3$ mixing ratios [ppb] from the stations are split up in day (yellow) and night (blue). The error bars indicate the standard deviation of the observational data from randomly sampled points (not all standard deviations are shown for visual purposes).





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
