# Peer review of "Evaluation of nitrogen oxides sources and sinks and ozone production in Colombia and surrounding areas"

_Atmospheric Chemistry and Physics, 2019_

## Referee Comment (RC1) · Anonymous Referee #1 · 3 Dec 2019

Summary of review:

This paper is well written from a technical point of view and the science is solid, but I recommend this current manuscript be split into two (or maybe even three) papers. It appears that the authors are combining several complementary papers into a single manuscript. One that is evaluating the sources and sinks of NO2 using WRF-Chem and OMI and its effects on ozone, one that is trying to resolve a discrepancy in PBL height, and perhaps even another on the role of lightning NOx in the Amazonia.

In my opinion, the authors have two options here: 1) either significantly shorten Section 4 & Discussion and add a stronger focus on the SCM model or 2) to exclude Section 5

entirely. I would prefer option 1, but I will leave that decision up to the authors. I also think option 1 better fits the scope of ACP.

I recommend publication, but only after the scope of the paper is narrowed.

Major comments:

Ln 260: I am uneasy with attributing the high model bias in the Amazonia to lightning NOx, primarily because the boundary of your domain is fairly close to this region and that climatological winds are generally from the east. I do not think it is correct to automatically assume lightning NOx is the reason for the discrepancy. I believe that boundary conditions could be playing a role here. Additionally, you may only be using <10 days of OMI data in the comparison (Figure 2), which is particularly an issue here since the NO2 measurements are near the lower limit of the OMI's capability. In general, a long discussion on the lightning NOx is not warranted because it is a small sample size near the instrument's detection limit. Please also revise the later parts of the manuscript when lightning NOx is discussed.

Ln 295-298: After looking at Figure 6, I am confused how the authors are implying that there is good agreement between the modeled NOx/O3/CO and surface monitors at any hour. Perhaps I am misinterpreting something, but if not, these sentences should be modified.

Ln 354 How is the boundary layer constrained in the single column model? This seems to be key information, but it is left out. In general, Section 5 is lacking specifics. As emphasized above, I think this could be a either a great follow-up paper or Section 4 should be shortened and this could be a larger focus of the paper.

Minor comments:

Ln 101: What initial conditions are used?

Ln 142: The words "on the large-scale" are probably unnecessary.
Ln ~190: Silvern et al., 2018 should at least be mentioned at this point in the manuscript. It suggests that the NO2/NO partitioning may not be good in the upper troposphere. The paper shows that NO2 in upper atmosphere is often too low in global models. This is important when calculating the AMF and could affect it significantly when NO2 is generally low such as the Amazonia region.

Silvern, R. F., Jacob, D. J., Travis, K. R., Sherwen, T., Evans, M. J., Cohen, R. C., Laughner, J. L., Hall, S. R., Ullmann, K., Crounse, J. D., Wennberg, P. O., Peischl, J., and Pollack, I. B.: Observed NO/NO2 Ratios in the Upper Troposphere Imply Errors in NO-NO2-O3 Cycling Kinetics or an Unaccounted NOx Reservoir, Geophys. Res. Lett., 45, 4466–4474, https://doi.org/10.1029/2018gl077728, 2018. 

Figure 2: Borders are hard to see. Perhaps change them to white? Also for clarity, perhaps change the values to % of the month instead of number of measurements.

Figure 4b: The units are unclear. Please clarify

Ln 250 & Ln 257: Insert the word "model" before "overestimation"

Ln 273: Is Figure 5d necessary? It does not seem to add any helpful information.

Ln 296: Should clarify to "morning rush hour"

Ln 359: Discussion section should be more concise.

Ln 376 - 390: I'm not sure how many overarching conclusions about lightning NOx can be made from this study. I suggest this paragraph be removed.

---

## Referee Comment (RC2) · Anonymous Referee #2 · 27 Jan 2020

In their paper, Barten et al. provide an evaluation of atmospheric chemistry over Colombia using WRF-Chem modelling evaluated with satellite and surface observations. The paper is very well written, contains a good and balanced set of references, has an appropriate length and amount of detail and I could not discover any obvious flaws. Colombia is a complex and interesting country with isolated regions and various climate zones where the dominant emissions may vary from anthropogenic, biomass burning to lightning. The approach is of wider interest, because similar studies may be conducted for other countries with limited air-quality monitoring networks. Figure 4-b is a central figure in the paper and starting point to discuss the different regions in more detail.

I am in favour of publishing this work, but I have four major general comments, provided below, which will require a major revision.

General comments:

1. The authors present only one month of simulations (January 2014) but the seasonal dependence of NOx and ozone is not discussed. There are good reasons to focus on January because it is the dry season, but the country experiences wet and dry seasons where the relative importance of sources of NO may change. The diurnal variability as well as the multi-annual variability in the satellite data are discussed, but the seasonality is missing. Would it be possible to extend the simulations to a couple of months to sample the yearly cycle in emissions? It would be interesting to present also the seasonality of the satellite (and surface) observations.

2. The authors show that the lightning source is the dominant source for 63% of the grid cells and is also the largest source in terms of total amount. Therefore lightning is a key aspect for Colombia, and, given also the major uncertainty in the modelling of this process, deserves special attention. The uncertainty in the lightning source is e.g. demonstrated by the adjustments made to the default settings of WRFchem, which scaled down lightning by a factor 20. However, the authors (if I understood correctly) have used only the clear-sky observations of OMI. It has been shown in several publications that lightning source estimates may be derived using the observations over high clouds. Because the resolution of WRF is comparable to OMI, it would be interesting to include a comparison between these cloud covered observations and WRF-Chem, to test the capability of the model to describe major thunderstorms. See for instance: Beirle, S., Huntrieser, H., and Wagner, T.: Direct satellite observation of lightning-produced NOx, Atmospheric Chemistry and Physics, 10, 10 965–10 986, https://doi.org/10.5194/acp-10-10965-2010, 2010. Pickering, K. E., Bucsela, E., Allen, D., Ring, A., Holzworth, R., and Krotkov, N.: Estimates of lightning NOx production based on OMI NO2 observations over the Gulf of Mexico, Journal of Geophysical Research: Atmospheres, 121, 8668–8691, https://doi.org/10.1002/2015JD024179, 2016.

3. The discussion of soil biogenic emissions is very limited. The regions where these emissions are dominant are identified. Therefore the comparison with OMI could be extended: are there indications that soil emissions are under (over) estimated? Where and by what amount?

4. The differences between the single-column model and WRF-Chem should be more clearly described. Why does WRF-Chem produce so much lower concentrations in the city?

Detailed comments:

Abstract, l15: "averaged difference of 0.02Åů10ˆ15". How significant is this number?

l19: "WRF-Chem was unable to capture NOx and CO urban". For which cities?

l80: "air quality in Colombia concerns are generally" Replace by "concerns about air quality in Colombia are generally"

l101: The spin-up time is very short? How is the model initialised?

l103: "in Appendix A we show how the selected study period can be deemed being representative for the baseline state of air quality in Colombia. " This claim is not very clear to me. Figure A1 shows the year-to-year mean variation in January. Why is it representative? I would like to see also a seasonal variation, e.g. linked to the wet and dry seasons. Only showing results for January is a weak point of the paper that should be better motivated. Also, a seasonal mean could be more representative because of the limited number of OMI pixels in a month, see Fig.2.

l107: " ... (ECMWF) .. meteorological boundary conditions." Is this the operational dataset or reanalysis?

l158: "data filtering recommendations by the QA4ECV": What is the filtering criterion for clouds? I assume cloud-covered pixels are not used?

l161: "limiting the quality of and which increases the uncertainty". Please reformulate.

l184: "mostly decreases in AMF". Please replace by "increases in the VCD" because not all readers will be familiar with the AMF concept. This remark applies to the whole section: please refer to VCD only.

l190: "This reflects a higher abundance of NO2 higher up in the troposphere". It is not fully clear to me how this can be concluded. The whole profile will be important. Are lightning emissions modelled higher in the atmosphere in WRF compared to the TM5-MP? Was this checked? Also, cloud-covered observations are removed from the OMI dataset by the filtering?!

l196: "In this research, we focus on tropospheric NO2 columns." This line can be removed.

Sec. 3.2 in-situ data: please provide information about the instruments used. Is the data publicly available?

Fig. 4a: Unit? Is this per grid cell, per square km, or something else?

Fig. 4b, showing a map of the dominant source, is a very useful plot!

l232-237: The numbers in the table are repeated in the text. This paragraph may be shortened therefore. I would suggest to add the % of land where the source is dominant (as given in the text) as extra column in the table.

l241: "very low VCD over Carribbean". What is the influence of the (free troposphere) boundary conditions?

l248: "northeastern part". Please provide a more detailed explanation where these high values come from.

l257: "Even though the overestimation is small in absolute terms". Could a possible bias in the satellite observations contribute to the difference observed?

l261: "This further confirms the finding that lightning NOx emissions are overestimated". Maybe this conclusion should be weakened. As explained by the authors

in l262-265, the comparison with OMI is only performed in OMI clear-sky conditions. Therefore the comparison may be biased.

l290: "These results confirm the application of the recalculated OMI data." It should be noted that the differences in mean, median, 90% confidence interval is not very large. So it is questionable if the confirmation is significant.

p14: I was wondering if the authors have analysed the precise location of the surface stations in Bogota? Closeness to major sources/roads could perhaps explain the difference with WRF-Chem?

l341: "nudging the concentrations of long-lived tracers such as O3, NOx and CO above the boundary layer using the CAMS data". Why not use the WRF-Chem data for this?

l342: "used the same emissions, including diurnal cycle, as in the WRF-Chem simulation." I find it conceptually difficult to understand why the SCM works better than WRF-Chem with such similar inputs. From the previous section I understood that the local (traffic) emissions at the surface stations are underestimated. But the SCM apparently uses the same emissions as WRF-Chem. Why are concentrations of NOx/CO so much higher in the SCM than in WRF-Chem? Which aspect of the SCM is responsible for this difference?

l367: "VCD analysis for January 2014 is representative for the NOx emissions for the larger study domain". I find this not well justified. The A1 plot only shows January. In particular the seasonal variability would be of interest while the study is limited to one year and January only.

l389: "still overestimates NOx emissions". Please explain why? The domain studied is a very active lightning region. What fraction of the global lightning total is expected to come from this area? E.g. the comparison with Miyazaki is quite close.

l400: "urband"

l415: "which explains part of the overestimation by WRF-Chem". It should be mentioned that there is a strong variability from year to year, most likely related to other reasons than changes in emissions which are expected to result in more gradual changes. See next line.

l423: "we found that WRF-Chem does not systematically underestimate urban VCDs". What is this statement based on? Do the authors refer to studies over other regions with the same setup?

l434-436:"overestimations of VCDs ... in contrast with ... there is an overestimation " ? Please reformulate.

l445; l457 : "indicating that all surface monitoring stations are located at or near busy roads", "EDGAR emissions integrated in a relatively simple Single Column Model, can represent the averaged diurnal cycles of O3, CO and NOx reasonably well.". This is confusing and should be explained more clearly. The SCM concentrations also match quite well in absolute amount the surface observations, which seems to suggest that EDGAR is OK and that the surface network is representative for larger areas?!

There is a repetition of concluding remarks when comparing the discussion and conclusion sections. This repetition should be removed. Maybe the two sections can be merged (and thereby shortened a bit)?

The close agreement of the mean/median is a bit over-emphasised to my taste. In the different source sectors there are major differences with compensating effects on the mean or total.

l489: "showed that the selected simulation period is representative for the baseline state of air quality in Colombia". Is this a statement for January only?

l497: "evaluate the impact of a modified representation of emissions based on the observed to WRF-Chem simulated CO mixing ratio". The description of the SCM, section 5, says: "and used the same emissions, including diurnal cycle, as in the WRF-Chem simulation". This is confusing. Please explain more clearly the setup of the

SCM and explain why e.g. the CO/NOx concentrations have increased substantially compared to WRF-Chem.

l507: "It may provide as a base for more local studies or the application towards future predictions". Please re-write.

---

## Author Comment (AC1) · 14 Apr 2020

**Author response to the referee comments to the paper by Barten et al.: Evaluation of nitrogen oxides sources and sinks and ozone production in Colombia and surrounding areas**

We would like to thank the two reviewers for their constructive feedback on the manuscript. Those reviews indicate that both reviewers support publication of this paper on an Colombian AQ assessment but both recommend also major modifications. We especially appreciate one of the reviewers' comment that Colombia is an interesting study area due to its various emission sources and isolated regions but also having the large diversity in biogeochemical regimes and complex topography. All comments greatly help in further improving the manuscript. Below, we address each comment individually. Referee comments are given in *italic*, author response are given in normal font, changes made to the text are given in blue. This document is finalized by a markdown version of the manuscript including all the changes.

**Review #1:**

**Summary of review:**

*This paper is well written from a technical point of view and the science is solid, but I recommend this current manuscript be split into two (or maybe even three) papers. It appears that the authors are combining several complementary papers into a single manuscript. One that is evaluating the sources and sinks of $NO_2$ using WRF-Chem and OMI and its effects on ozone, one that is trying to resolve a discrepancy in PBL height, and perhaps even another on the role of lightning $NO_x$ in the Amazonia. In my opinion, the authors have two options here: 1) either significantly shorten Section 4 & Discussion and add a stronger focus on the SCM model or 2) to exclude Section 5 entirely. I would prefer option 1, but I will leave that decision up to the authors. I also think option 1 better fits the scope of ACP. I recommend publication, but only after the scope of the paper is narrowed.*

We thank anonymous referee #1 for the critical review and constructive comments. The idea of the paper was to apply and evaluate WRF-Chem by comparison with remote sensing and few available in situ observations to study air quality in regions/countries with a limited in situ network such as Colombia. We decided to limit the presented study to identification of some of the main issues in this assessment of Colombian AQ such as the lightning $NO_x$ parametrization of WRF-Chem but also how the model performs regarding simulations of observed sub-grid scale levels of urban pollution. The proposed more detailed analysis of $NO_x$ lightning, should be focus in follow-up studies since, as also indicated by reviewer, this analysis would be more than sufficient for another dedicated study. According to the reviewer, the same holds for evaluation of the boundary layer dynamics relative to the (mis)representation of emissions introducing application of the SCM simulations. Based on the reviewers comments, we have considered to indeed completely remove Section 5 but have now decided to handle the raised comments by 1) substantially shortening this section (5) on the SCM results, 2) to be less quantitative regarding the SCM results and, 3) merging these results with section 4.3. So, we still would like to make this specific point in this paper on a misrepresentation of emissions relative to the role of boundary layer dynamics in explaining discrepancies between simulated and observed concentrations. But we revised the text to stress more explicitly that the SCM exercise was not to propose a new modelling strategy with an improved representation of BL dynamics, but mainly aimed to conduct a sensitivity experiment to complement the evaluation of WRF-Chem simulated urban area pollutant concentrations. Consistent with these changes in Section 4 & 5, the abstract, discussion and conclusions have been modified accordingly.

***Major comments:***

*Ln 260: I am uneasy with attributing the high model bias in the Amazonia to lightning $NO_x$, primarily because the boundary of your domain is fairly close to this region and that climatological winds are generally from the east. I do not think it is correct to automatically assume lightning $NO_x$ is the reason for the discrepancy. I believe that boundary conditions could be playing a role here. Additionally, you may only be using <10 days of OMI data in the comparison (Figure 2), which is particularly an issue here since the $NO_2$ measurements are near the lower limit of the OMI's capability. In general, a long discussion on the lightning NOx is not warranted because it is a small sample size near the instrument's detection limit. Please also revise the later parts of the manuscript when lightning $NO_x$ is discussed.*

Also based on the comments by Reviewer #2 and re-reading the text, we realize that we might have been to explicit in our statements regarding overestimation of the lightning $NO_x$ emissions. We have addressed the issue that only a small number of measurements are available over these regions although we have already taken January 2014 as simulation month, but this does not outweigh the statements on the overestimation of the emissions. Based on the reviewers comments, we have once more again carefully evaluated especially the model's eastern boundary conditions, but in general, very little NOx is advected over the domain boundary (also visible in the eastern most grid cells in fig. 5a). We agree that OMI is operating here near the detection limit. We have updated the manuscript in many sections, mostly reducing the statements on the overestimation of lightning $NO_x$ emissions.

*Ln 295-298: After looking at Figure 6, I am confused how the authors are implying that there is good agreement between the modeled $NO_x/O_3/CO$ and surface monitors at any hour. Perhaps I am misinterpreting something, but if not, these sentences should be modified.*

The text was referring to Figure C1 in the Appendix. The point we wanted to stress is that WRF-Chem generally represents the lower limit in observed $NO_x$ mixing ratios and the upper limit of observed $O_3$ mixing ratios during daytime. However, WRF is not able to simulate the high $NO_x$ ratios found during rush hour or the low nocturnal $O_3$ (< 5 ppb) mixing ratios of the individual stations. We have modified the text such that it should be clear that we are discussing those upper and lower limits found in the individual (and not the average diurnal cycle) measurements.

*Ln 354 How is the boundary layer constrained in the single column model? This seems to be key information, but it is left out. In general, Section 5 is lacking specifics. As emphasized above, I think this could be a either a great follow-up paper or Section 4 should be shortened and this could be a larger focus of the paper.*

Boundary layer (BL) dynamics in the SCM are calculated in a similar manner compared to WRF-Chem in online calculations involving surface and boundary layer exchange of momentum, energy, moisture (and tracers) but then using different BL schemes (YSU in WRF-Chem, ECHAM4 climate model BL scheme in SCM). The details can also be found in previous references on SCM studies (e.g., Ganzeveld et al., 2002b). We might have though not been clear enough that, in contrast to offline models that prescribe the BL depth, the SCM calculates this BL dynamics in an online set-up similar to WRF-Chem. We have further stressed this essential feature: e.g., Section 4.3 "The SCM simulates online, similar to WRF-Chem, atmospheric chemistry processes, including anthropogenic and natural emissions, gas-phase chemistry, wet and dry deposition and turbulent and convective tracer transport as a function of meteorological and hydrological drivers, surface cover, and land use properties (Ganzeveld et al., 2002b, 2008)". However, rather providing the details of BL dynamics in the SCM (and WRF-Chem) in a more elaborated Section 5, we decided to strongly shorten and merge this section with Section 4.3. Furthermore, also being less quantitative we refrain from any claim that the SCM performs better regarding BL dynamics and which is also not possible not having BL depth/structure measurements. If such measurements would be available then it would definitely be worthwhile to further investigate the representation of urban boundary layer dynamics in the SCM and WRF-Chem also motivated by the presented simulated large differences in pollutant diurnal cycles. Throughout the manuscript we have modified the description of the purpose and main findings of the supporting experiments with the SCM.

***Minor comments:***

*Ln 101: What initial conditions are used?*

The ECMWF ERA-Interim and CAMS data products used for the boundary conditions are also used for the initial conditions. This is updated in the text.

*Ln 142: The words "on the large-scale" are probably unnecessary.*

We have removed "large scale"

*Ln _190: Silvern et al., 2018 should at least be mentioned at this point in the manuscript. It suggests that the $NO_2/NO$ partitioning may not be good in the upper troposphere. The paper shows that $NO_2$ in upper atmosphere is often too low in global models. This is important when calculating the AMF and could affect it significantly when $NO_2$ is generally low such as the Amazonia region. Silvern, R. F., Jacob, D. J., Travis, K. R., Sherwen, T., Evans, M. J., Cohen, R. C., Laughner, J. L., Hall, S. R., Ullmann, K., Crounse, J. D., Wennberg, P. O., Peischl, J., and Pollack, I. B.: Observed NO/NO2 Ratios in the Upper Troposphere Imply Errors in NO-NO2-O3 Cycling Kinetics or an Unaccounted NOx Reservoir, Geophys. Res. Lett., 45, 4466–4474, https://doi.org/10.1029/2018gl077728, 2018.âˇA´C*

This can indeed be an explaining factor. We have added the reference.

*Figure 2: Borders are hard to see. Perhaps change them to white? Also for clarity, perhaps change the values to % of the month instead of number of measurements.*

We have changed the values and borders.

*Figure 4b: The units are unclear. Please clarify*

The figure indicates the spatial distribution of the dominant emission source. The saturation of the color indicates the % of the total flux coming from the dominant emission source going up to 100% for the darkest colors. We have updated the caption to make it more clear.

*Ln 250 & Ln 257: Insert the word "model" before "overestimation"*

Added model twice.

*Ln 273: Is Figure 5d necessary? It does not seem to add any helpful information.*

We have removed figure 5d, also combining the colorbar of 5a and 5b.

*Ln 296: Should clarify to "morning rush hour"*

Added morning.

*Ln 359: Discussion section should be more concise.*

We have reduced the information in the discussion. Especially also removing a section where we discuss in detail the representativeness of EDGAR anthropogenic emissions above Bogota (Colombia) and Caracas (Venezuela) due to a decrease in economic activity. This analysis extra was not directly supporting the main goal/message of the manuscript.

*Ln 376 - 390: I'm not sure how many overarching conclusions about lightning $NO_x$ can be made from this study. I suggest this paragraph be removed.*

We realize that these points have indeed been addressed before (methods/results/beginning of discussion) and come back later in the discussion. We have removed this section for the large part (thereby making the discussion section more concise) and moved Further attention is required no only regarding the lightning $NO_x$ parametrization scheme, but also the model representation of convection and clouds, in follow-up studies on atmospheric $NO_x$ over Colombia, or other regions where lightning is a dominant source of $NO_x$. further down in the discussion.

**Review #2:**

*In their paper, Barten et al. provide an evaluation of atmospheric chemistry over Colombia using WRF-Chem modelling evaluated with satellite and surface observations. The paper is very well written, contains a good and balanced set of references, has an appropriate length and amount of detail and I could not discover any obvious flaws. Colombia is a complex and interesting country with isolated regions and various climate zones where the dominant emissions may vary from anthropogenic, biomass burning to lightning. The approach is of wider interest, because similar studies may be conducted for other countries with limited air-quality monitoring networks. Figure 4-b is a central figure in the paper and starting point to discuss the different regions in more detail. I am in favour of publishing this work, but I have four major general comments, provided below, which will require a major revision.*

We thank anonymous referee #2 for her/his review of our manuscript and the very constructive feedback. We already mentioned previously that we really acknowledge the comment what makes Colombia being an interesting study area regarding meteorology and AQ but also appreciate the comment by this reviewer that the selected approach is of wider interest.

**General comments:**

*1. The authors present only one month of simulations (January 2014) but the seasonal dependence of $NO_x$ and ozone is not discussed. There are good reasons to focus on January because it is the dry season, but the country experiences wet and dry seasons where the relative importance of sources of NO may change. The diurnal variability as well as the multi-annual variability in the satellite data are discussed, but the seasonality is missing. Would it be possible to extend the simulations to a couple of months to sample the yearly cycle in emissions? It would be interesting to present also the seasonality of the satellite (and surface) observations.*

One of the goals of this manuscript was to present a concise method to study air quality in complex regions like Colombia with a limited monitoring network in rural areas. The goal was not necessarily to study the capability of WRF-Chem to simulate these $NO_2$ (and $O_3$ urban) column/mixing ratios over the whole year. We decided to focus on January (2014) mainly because of the larger availability of remote sensing data with the generally reduced cloud cover. We have also added the yearly trend to indicate some features relevant to Colombia (El Nino) and to put it in perspective. We appreciate the reviewers comment and agree that a seasonal analysis of the OMI data would finalize this, without the need to conduct extra WRF-Chem simulations. We have added a subfigure, to figure A1, that shows the seasonality of the $NO_2$ columns above dominant emission sources including the spread (standard deviation) in the years 2005-2019. We discuss this figure in the Discussion section.

*2. The authors show that the lightning source is the dominant source for 63% of the grid cells and is also the largest source in terms of total amount. Therefore lightning is a key aspect for Colombia, and, given also the major uncertainty in the modelling of this process, deserves special attention. The uncertainty in the lightning source is e.g. demonstrated by the adjustments made to the default settings of WRFchem, which scaled down lightning by a factor 20. However, the authors (if I understood correctly) have used only the clear-sky observations of OMI. It has been shown in several publications that lightning source estimates may be derived using the observations over high clouds. Because the resolution of WRF is comparable to OMI, it would be interesting to include a comparison between these cloud covered observations and WRF-Chem, to test the capability of the model to describe major thunderstorms. See for instance: Beirle, S., Huntrieser, H., and Wagner, T.: Direct satellite observation of lightning-produced NOx, Atmospheric Chemistry and Physics, 10, 10 965–10 986, https://doi.org/10.5194/acp-10-10965-2010, 2010. Pickering, K. E., Bucsela, E., Allen, D., Ring, A., Holzworth, R., and Krotkov, N.: Estimates of lightning NOx production based on OMI NO2 observations over the Gulf of Mexico, Journal of Geophysical Research: Atmospheres, 121, 8668–8691, https://doi.org/10.1002/2015JD024179, 2016.*

It was not necessarily the goal of the paper to put more emphasis on the representation of lightning $NO_x$ emissions. However, in analysing the results (and from other literature) it became clear that this is indeed a very uncertain process in regional air quality modelling and the region focus of our study. It was

also not the goal to arrive at more quantitative estimates of (lightning $NO_x$) emissions but rather use the satellite data, besides the limited in-situ observations, to study regional air quality and define the different source regions. We concede to the comment of Referee #1 that the paper is already potentially too 'broad'. We also agree that an analysis like you mention is a very insightful tool to look at the lightning $NO_x$ emissions in more depth and suggest that it would be a very interesting follow-up research. We have updated the manuscript with the following statement in the discussion: This study does not aim to provide comprehensive estimates of any of the emission sources using OMI data. Rather, we show the potential use of satellite data in a region with a limited air quality monitoring network in determining the regional scale air quality and $NO_x$ source regions. The use of cloud covered OMI observations to get a more comprehensive estimate of lightning $NO_x$ emissions (Beirle et al., 2010; Pickering et al., 2016) would make a very interesting follow up study.

*3. The discussion of soil biogenic emissions is very limited. The regions where these emissions are dominant are identified. Therefore the comparison with OMI could be extended: are there indications that soil emissions are under (over) estimated? Where and by what amount?*

We realize that soil emissions can potentially be underestimated by MEGAN, especially in regions with land management practises such as fertilizer application also after we found out in a recent other application of WRF-Chem for Europe (Visser et al., 2019) that this contribution to soil NO emissions is not considered in MEGAN/WRF-Chem. However, the OMI-WRF comparison indicates no clear under- or overestimation of biogenic emissions. In the discussion section we have added a comparison with the global chemistry-climate model EMAC. The MEGAN biogenic emissions in this study seem to be in line with the estimates by EMAC for the same region. Also, the contribution of fertilizer to the total soil NO flux seems to be very limited (~1.3%).

*4. The differences between the single-column model and WRF-Chem should be more clearly described. Why does WRF-Chem produce so much lower concentrations in the city?*

The differences appear to be mostly associated with the representation of the nocturnal inversion layer with the SCM simulations being nudged (e.g., Ganzeveld et al., 2006) with the WRF-Chem meteorology resulting a shallower nocturnal inversion layers compared to WRF-Chem. It is both the depth of the nocturnal inversion layer but also the timing of the onset of turbulent mixing relative to the onset of the morning rush hour and associated emissions that explains the large differences. But (see also previous comments), we have substantially shortened that section also aiming to present this more as a sensitivity analysis also given the fact that we cannot proof which of the different simulations are more realistic not having detailed BL dynamics observations. Throughout the manuscript we have modified the description of the purpose and main findings of the supporting experiments with the SCM.

**Detailed comments:**

*Abstract, l15: "averaged difference of 0.02·u10ˆ15". How significant is this number?*

We have included the 90% Confidence Interval to the text to indicate the spread around the mean. We have also changed it to a mean bias (WRF-Chem minus OMI) to make it more clear that it is about the bias between OMI and WRF.

*l19: "WRF-Chem was unable to capture $NO_x$ and CO urban". For which cities?*

The urban comparison was done for all four cities (Cali, Bogota, Medellin, Bucaramanga) with air quality monitoring data. Therefore, the statement is in general for all Colombian cities with air quality monitoring data. Because the comparison showed similar results for the different cities we decided to only show the results for Bogota since they are most robust due to the presence of multiple measurement stations.

*l80: "air quality in Colombia concerns are generally" Replace by "concerns about air quality in Colombia are generally"*

Updated

*l101: The spin-up time is very short? How is the model initialised?*

The model is initialized with ECMWF ERA-Interim (meteorology) and CAMS (chemistry). For the surface mixing ratios, the signal of the initialization seems to be gone after 24 hours due to the local emissions and advection.

*l103: "in Appendix A we show how the selected study period can be deemed being representative for the baseline state of air quality in Colombia. " This claim is not very clear to me. Figure A1 shows the year-to-year mean variation in January. Why is it representative? I would like to see also a seasonal variation, e.g. linked to the wet and dry seasons. Only showing results for January is a weak point of the paper that should be better motivated. Also, a seasonal mean could be more representative because of the limited number of OMI pixels in a month, see Fig.2.*

See reply to general comment #1

*l107: " ... (ECMWF) .. meteorological boundary conditions." Is this the operational dataset or reanalysis?*

As stated above, this is the ERA-Interim product. We have updated this accordingly in the text.

*l158: "data filtering recommendations by the QA4ECV": What is the filtering criterion for clouds? I assume cloud-covered pixels are not used?*

The criterion for clouds is: cloud radiance fraction > 0.5. We have added this to the text. This is by far the strictest criterion for the selected study period.

*l161: "limiting the quality of and which increases the uncertainty". Please reformulate.*

We have changed the sentence to This limits the quality of the measurements and which increases the uncertainty in the averaged tropospheric $NO_2$ column.

*l184: "mostly decreases in AMF". Please replace by "increases in the VCD" because not all readers will be familiar with the AMF concept. This remark applies to the whole section: please refer to VCD only.*

We have updated this in the manuscript.

*l190: "This reflects a higher abundance of $NO_2$ higher up in the troposphere". It is not fully clear to me how this can be concluded. The whole profile will be important. Are lightning emissions modelled higher in the atmosphere in WRF compared to the TM5-MP? Was this checked? Also, cloud-covered observations are removed from the OMI dataset by the filtering?!*

In this specific case, we have checked the individual and average TM5/WRF-Chem profiles with valid observations (not filtered out by clouds).

*l196: "In this research, we focus on tropospheric $NO_2$ columns." This line can be removed.*

Removed.

*Sec. 3.2 in-situ data: please provide information about the instruments used. Is the data publicly available?*

The data and information on the instruments is publicly available at http://sisaire.ideam.gov.co/ideam-sisaire-web/

*Fig. 4a: Unit? Is this per grid cell, per square km, or something else?*

The unit was initially per grid cell as indicated in the caption. We realize that a flux per unit area (square km) is easier to interpret. We have changed this in the figure, text and caption accordingly.

*l232-237: The numbers in the table are repeated in the text. This paragraph may be shortened therefore. I would suggest to add the % of land where the source is dominant (as given in the text) as extra column in the table.*

We have reduced the text and remove the numbers from this paragraph.

*l241: "very low VCD over Caribbean". What is the influence of the (free troposphere) boundary conditions?*

We are analysing here the complete modeling system (so WRF-Chem including emission inventories and the boundary conditions). The prevailing winds are easterlies. So this is not the region where we would expect the largest influence of the boundary conditions.

*l248: "northeastern part". Please provide a more detailed explanation where these high values come from.*

These high VCDs are mostly caused by a lot of $NO_2$ close to the surface in the CAMS boundary conditions advected into the WRF-Chem domain by easterly winds. We have updated this in the manuscript.

*l257: "Even though the overestimation is small in absolute terms". Could a possible bias in the satellite observations contribute to the difference observed?*

We realize that OMI is operating here at its detection limit and that only clear-sky observations are considered. We have updated the text on Lightning $NO_x$ in several sections also based on Review #1.

*l261: "This further confirms the finding that lightning $NO_x$ emissions are overestimated". Maybe this conclusion should be weakened. As explained by the authors in l262-265, the comparison with OMI is only performed in OMI clear-sky conditions. Therefore the comparison may be biased.*

See previous comment.

*l290: "These results confirm the application of the recalculated OMI data." It should be noted that the differences in mean, median, 90% confidence interval is not very large. So it is questionable if the confirmation is significant.*

Since this was also not the main goal of the procedure we have decided to leave out the analysis on the reduction of the bias using the recalculated OMI columns. We have also revised this in earlier and later parts of the manuscript.

*p14: I was wondering if the authors have analysed the precise location of the surface stations in Bogota? Closeness to major sources/roads could perhaps explain the difference with WRF-Chem?*

We have done this, we think that all the surface stations should give a representative signal for the (Bogota) urban mixing ratios. The spread of the mixing ratios measured at the stations can be seen in Figure C1 (Appendix).

*l341: "nudging the concentrations of long-lived tracers such as $O_3$, $NO_x$ and CO above the boundary layer using the CAMS data". Why not use the WRF-Chem data for this?*

Based on this comment we have conducted once more again the experiments with the SCM applying, instead of the CAMS data, the WRF-Chem simulated free tropospheric mixing ratios of $O_3$, $NO_x$ and CO. Also, the initially indicated numbers were not properly updated to the selected values of the actual simulation set-up presented in the paper. The new setup and results are now included in the revised document in Figure 6 and presented in the modified Section 4.3.

*l342: "used the same emissions, including diurnal cycle, as in the WRF-Chem simulation." I find it conceptually difficult to understand why the SCM works better than WRF-Chem with such similar inputs. From the previous section I understood that the local (traffic) emissions at the surface stations are underestimated. But the SCM apparently uses the same emissions as WRF-Chem. Why are*

*concentrations of NOx/CO so much higher in the SCM than in WRF-Chem? Which aspect of the SCM is responsible for this difference?*

See previous reply regarding the differences between the SCM and WRF-Chem simulated diurnal cycles in inversion/boundary layer depth

*l367: "VCD analysis for January 2014 is representative for the NOx emissions for the larger study domain". I find this not well justified. The A1 plot only shows January. In particular the seasonal variability would be of interest while the study is limited to one year and January only.*

See reply to general comment #1

*l389: "still overestimates $NO_x$ emissions". Please explain why? The domain studied is a very active lightning region. What fraction of the global lightning total is expected to come from this area? E.g. the comparison with Miyazaki is quite close.*

The domain in Miyazaki et al. (2014) covers a larger domain of South-America including the complete Amazon region where lightning is very active. Also based on other comments, we have restructured and rewritten results, discussion and conclusions on lightning $NO_x$ emissions.

*l400: "urband"*

Changed to urban.

*l415: "which explains part of the overestimation by WRF-Chem". It should be mentioned that there is a strong variability from year to year, most likely related to other reasons than changes in emissions which are expected to result in more gradual changes. See next line.*

We have chosen to remove this section entirely also based on Reviewer #1 comment to reduce the content in the Discussion. We see that this section made the manuscript more 'broad' then intended especially since it does not support the main goal of the manuscript.

*l423: "we found that WRF-Chem does not systematically underestimate urban VCDs". What is this statement based on? Do the authors refer to studies over other regions with the same setup?*

We have chosen to remove this section entirely also based on Reviewer #1 comment to reduce the content in the Discussion. We see that this section made the manuscript more 'broad' then intended especially since it does not support the main goal of the manuscript.
The statement was based on the OMI-WRF comparison specifically for urban regions (similar to the OMI-WRF comparison for lightning and biomass burning regions, but not explicitly mentioned in the manuscript)

*l434-436:"overestimations of VCDs ... in contrast with ... there is an overestimation " ? Please reformulate.*

We have changed the second 'overestimation' to underestimation. We have also added modeled to make it easier to read.

*l445; l457 : "indicating that all surface monitoring stations are located at or near busy roads", "EDGAR emissions integrated in a relatively simple Single Column Model, can represent the averaged diurnal cycles of $O_3$, CO and $NO_x$ reasonably well.". This is confusing and should be explained more clearly. The SCM concentrations also match quite well in absolute amount the surface observations, which seems to suggest that EDGAR is OK and that the surface network is representative for larger areas?!*

We think that all the surface stations (especially the averaged diurnal cycle) should give a representative signal for the (Bogota) urban mixing ratios. We have removed the (incorrect) statement that all measurement stations are located near busy roads but still indicated that single measurements do not rarely exceed 150 ppb $NO_x$ (to indicate the spatial variability).

*There is a repetition of concluding remarks when comparing the discussion and conclusion sections. This repetition should be removed. Maybe the two sections can be merged (and thereby shortened a bit)?*

Also based on the remarks by the other reviewer we have strongly reduced the length of the discussion section and also tried to minimize the repetition of concluding remarks.

*The close agreement of the mean/median is a bit over-emphasised to my taste. In the different source sectors there are major differences with compensating effects on the mean or total.*

In both the abstract and conclusion we have changed the text such that we indicate the mean bias, but shortly after indicate that there are over- and underestimations in regions with lightning and biomass burning respectively.

*l489: "showed that the selected simulation period is representative for the baseline state of air quality in Colombia". Is this a statement for January only?*

See reply to general comment #1. We have updated this statement also to include the seasonal variability.

*l497: "evaluate the impact of a modified representation of emissions based on the observed to WRF-Chem simulated CO mixing ratio". The description of the SCM, section 5, says: "and used the same emissions, including diurnal cycle, as in the WRFChem simulation". This is confusing. Please explain more clearly the setup of the SCM and explain why e.g. the CO/NOₓ concentrations have increased substantially compared to WRF-Chem.*

Initially, the hypothesis was that a modified representation of the emissions (based on the observed to WRF-Chem simulated CO mixing ratio) would lead to a better agreement between model and observations. However, it appeared that the choice of the modelling system alone (with different advection and mixing conditions) already resulted in a much better agreement as indicated in l497: "This was actually achieved using the EDGAR emissions as also applied in WRF-Chem and mainly due to especially a different representation of advection and (nocturnal) mixing conditions.". For clarity, we have removed "to evaluate the impact of a modified representation of emissions based on the observed to WRF-Chem simulated CO mixing ratio"

[revised manuscript text omitted]

570 ity in Colombia  and which is essential to apply the presented combined modeling and measurement approach also to assess how air quality will further change due to future industrialization and land use changes.

575 *Code and data availability.* OMI data, in situ data and WRF-Chem output are available upon request as well as scripts to recalculate the tropospheric AMF.

*Author contributions.* JGMB and LNG designed the experiment. JGMB performed the WRF-Chem simulations. LNG performed the single column model simulations. JGMB performed the analysis and wrote the manuscript, with contributions from all co-authors.

*Competing interests.* The authors declare no competing interests.

580 *Acknowledgements.* The authors acknowledge Oscar Julian Guerrero Molina from the Instituto de Hidrología, Meteorología y Estudios Ambientales (IDEAM) for providing us with air quality monitoring data. We acknowledge WRF-Chem developers and emission inventory (EDGAR, MEGAN, GFED) developers. We also acknowledge the QA4ECV consortium for making OMI $NO_2$ data publicly available. We thank Folkert Boersma for his input on the OMI analysis.

**Appendix A:  Long term analysis of OMI VCD**

[Figure]

**Figure A1.** (a) January monthly averaged NO₂ vertical column densities [$10^{15}$ molecules cm⁻²] retrieved from OMI for 2005-2019 for the whole domain (black), regions with dominating anthropogenic (red), biogenic (green), biomass burning (yellow) and lightning (blue) emissions. The grey vertical bar highlights the WRF-Chem simulated year 2014. The red bars indicate El Niño years (2005, 2007, 2010, 2015, 2016, 2019). (b) Monthly averaged OMI NO₂ vertical column densities [$10^{15}$ molecules cm⁻²] for 2005-2019. The shadings indicate +/- 1 standard deviation.

**Table B1.** Available air quality monitoring stations including city, location and measured compounds.

| Station name | City | Latitude | Longitude | CO | NO | NO$_2$ | O$_3$ |
|---|---|---|---|---|---|---|---|
| Pance | Cali | 3.305 | -76.533 | | | | ✓ |
| Universidad del Valle | Cali | 3.378 | -76.534 | | | ✓ | ✓ |
| Compartir | Cali | 3.428 | -76.467 | | | | ✓ |
| C. Alto Rendimiento | Bogotá | 4.658 | -74.084 | ✓ | | | ✓ |
| Carvajal - Sevillana | Bogotá | 4.596 | -74.149 | ✓ | | | ✓ |
| Fontibon | Bogotá | 4.670 | -74.142 | ✓ | | | ✓ |
| Kennedy | Bogotá | 4.625 | -74.161 | ✓ | ✓ | ✓ | |
| Las Ferias | Bogotá | 4.691 | -74.083 | ✓ | | | ✓ |
| MinAmbiente | Bogotá | 4.626 | -74.067 | | | | ✓ |
| Puente Aranda | Bogotá | 4.632 | -74.118 | ✓ | ✓ | ✓ | ✓ |
| San Christobal | Bogotá | 4.573 | -74.084 | | | | ✓ |
| Tunal | Bogotá | 4.576 | -74.131 | ✓ | ✓ | ✓ | ✓ |
| Guaymaral | Bogotá | 4.784 | -74.044 | | ✓ | ✓ | ✓ |
| Suba | Bogotá | 4.761 | -74.094 | | ✓ | ✓ | ✓ |
| Usaquen | Bogotá | 4.710 | -74.030 | | | | ✓ |
| CAL-Corp. Lasallista | Medellín | 6.102 | -75.642 | | | | ✓ |
| ITA-Casa Justicia | Medellín | 6.188 | -75.601 | | ✓ | ✓ | |
| ITA-Col. Concejo | Medellín | 6.171 | -75.648 | | | | ✓ |
| MED-Politecnico JIC | Medellín | 6.212 | -75.581 | | ✓ | ✓ | |
| MED-Politecnico JIC (S) | Medellín | 6.212 | -75.581 | | ✓ | | |
| BEL-U.S. Buenaventura | Medellín | 6.331 | -75.569 | | ✓ | ✓ | ✓ |
| MED-Museo Antioquia | Medellín | 6.253 | -75.570 | ✓ | | | |
| MED-UN Fac. Minas | Medellín | 6.274 | -75.593 | | ✓ | ✓ | |
| MED-UN Nucleo Volador | Medellín | 6.266 | -75.580 | | ✓ | ✓ | ✓ |
| MED-Univ. Medellín | Medellín | 6.256 | -75.559 | | | | ✓ |
| MED-Villahermosa | Medellín | 6.256 | -75.559 | | | | ✓ |
| BAR-Parque Las Aguas | Medellín | 6.409 | -75.417 | | | | ✓ |
| Cabecera | Bucaramanga | 7.113 | -73.111 | ✓ | | | |
| Centro | Bucaramanga | 7.119 | -73.127 | ✓ | ✓ | ✓ | |
| Ciudadela | Bucaramanga | 7.106 | -73.124 | ✓ | | | |

**Appendix C: O₃ and NOₓ mixing ratios in Bogotá for January 2014**

[revised manuscript text omitted]

---

## Author Response (AR2)

**Author response to minor revisions request from Anonymous Referee #1**

We greatly appreciate the effort of both anonymous reviewers to read the revised manuscript and the minor comments provided by one of the reviewers to further improve the manuscript. Below, we address each comment individually. Referee comments are given in *italic*, author response are given in normal font, changes made to the text are given in blue. This document is finalized by a markdown version of the manuscript including all the changes.

*Thank you for these revisions. Indeed a lot of work has gone into improving this; I appreciate it. I have one further minor revision request, and the rest are technical corrections.*

We have added the following statement in the Acknowledgement section to express our appreciation to both the reviewers: We greatly appreciate the two anonymous reviewers for their critical and constructive comments. Their effort has contributed to major improvements of this manuscript.

*Minor Request:*

*-Thank you for clarifying the SCM. Now that I have a somewhat better understanding of it and it's comparison to WRF-Chem, I have one additional request. I think it's necessary to include an intercomparison of the modeled PBLs in Figure 6. There should be a fifth panel (or perhaps the NO panel can be removed). In this additional panel, you should display the diurnal cycle of the WRF-Chem PBL and the SCM PBL (and ideally the ERA5 PBL too if possible) for the same time frame.*

We have included the comparison of WRF-Chem, SCM and ERA5 PBLH's in the manuscript and removed the NO panel. We have made minor adjustments to the result section where we discuss and refer to the (sub-) figure and updated the figure caption. We have updated the text where we discuss the results of PBLH to:
Figure 7d shows a comparison of the SCM, WRF-Chem simulated- as well as the ERA5 reanalysis boundary layer height for the grid resembling the location of Bogota. The SCM is showing a substantially deeper daytime maximum boundary layer with more day-to-day variation compared to WRF-Chem and ERA5 reanalysis data. The SCM also simulates a relatively fast afternoon transition to suppressed nocturnal mixing conditions reflected by a nocturnal inversion layer which agrees well with the ERA5 boundary layer height being shallower than that simulated by WRF-Chem. Interestingly, the SCM simulation results in a better representation of the observed diurnal cycle of urban pollutant mixing ratios, especially regarding the observed early morning maximum CO and $NO_x$ and minimum $O_3$ concentrations, without requiring the hypothesized enhancement in emissions. This stresses that, besides application of higher-resolution emission inventories and model experiments, the diurnal cycle in boundary layer dynamics (and advection) should be critically evaluated in models such as WRF-Chem which, however, would then also require urban boundary layer structure measurements.

*Technical Requests:*

*-Abstract: The abstract is too long as currently written. I understand that there is no word limit for this journal, but this Abstract could be more concise (ideally <400 words). Some specific suggestions would be to remove Line 9 - 17 starting with sentence "The comparison... mean column". Also, the last 8 lines, Lines 27 - 34 can be shortened significantly.*

Thank you for your suggestions to reduce the abstract. We have reduced the abstract from 34 lines to 25 lines especially removing some of the technical model set-up statements. Also in the last 8 lines we have removed the line "Futhermore, this study ... mixing ratios" since it has been implicitly mentioned before when discussing the SCM simulations.

*-Abstract: With all of that said, there is one missing item in the Abstract and it is the mention of anthropogenic NOx in comparison to the other sources. It would be helpful to include this near Line 7 (similar to how it is mentioned in the Conclusions near Line 430).*

We have added the anthropogenic (but also the soil biogenic to 'close' the budget) and rewritten the sentence to: The model indicates the largest contribution by lightning emissions (1258 Gg N yr$^{-1}$), even after already significantly reducing the emissions, followed by anthropogenic (933 Gg N yr$^{-1}$), soil biogenic (187 Gg N yr$^{-1}$) and biomass burning emissions (104 Gg N yr$^{-1}$).

*-Line 123: Comma between "areas" and "the"*

Added ,.

*-Line 168: "dependens" --> "depends"*

Changed to depends.

*-Line 201: I am a bit uneasy that the lower limit filter is 1 pixel (i.e., any location with 2+ pixels are included). It seems from Figure 2, that the limit could be 5 and a large fraction of the area would still be valid. Is there a reason why 1 pixel is the lower limit. For example is a particular city/location excluded if the lower filter limit is 5 instead? If so, please mention this in the text.*

In an earlier stage of the study we have looked at the sensitivity to increasing/decreasing this number of available OMI measurements (from ≥2 to ≥4 to ≥6). It indicated that the sensitivity to the frequency distributions (figure 5d and 5e) is low and the mean/median and 90% CI remain relatively the same. However, by applying the 2+ criterion (as in the manuscript) we omit 10% of the total surface area of the domain. By applying the 4+ criterion this would become 20% and by applying the 6+ criterion this would become 34%. This would mainly go at the cost of a part of the Amazon rainforest, but also some of the cities in the mountainous regions. We agree that a 4+ or 6+ criterion would mean a better estimate of the monthly mean NO$_2$ column. However, we choose to stick with the 2+ criterion because the focus of the study was not to identify an accurate estimate of the monthly mean NO$_2$ columns in January, but more on their spatial distributions,

comparison with WRF-Chem and link with the four distinct emission regimes which benefits from a larger surface area cover.

Please note that after the interactive discussion stage we have updated Figure 2 from the exact number of available measurements to the % of measurements available during the simulation period (January 2014). So 2 pixels corresponds to ~5%.

*-Figure 3. Minor suggestion, but could be helpful to include the phase "AMF change" directly on the top center of the panel a, and the phrase "Column NO2 change" directly on the top center of panel b.*

We have updated the figure with the corresponding titles.

*-Figure 4b. Thanks for addressing this figure caption, but I don't think it's perfectly clear yet. I see the dark orange color in the center of this panel, yet this color is not in the colorbar (biomass burning is closest but appears to be a light orange). The reason I am confused is that the dark orange, at first glance, seems to represent a combination of biomass burning and anthropogenic. Yet, your explanation makes it seem like the dark orange color is only biomass burning. Either way, dark orange is very hard to differentiate from red. Perhaps a way to remedy this, would be to make the biomass burning a brighter shade of yellow, so that it is clearly different from the other colors.*

Thank you for your suggestion to improve the image. The lighter yellow colors are in their turn (to our perception) hard to distinguish from the lighter green colors (which are next to eachother in the Orinoco region). The most optimal combination (to our perception) is to change the biomass burning colors to purple. We have updated the figure, but also updated the (new) figure 8 to keep the colors consistent throughout the manuscript.

*-Figure 5. Please write the word "NO2" somewhere on this figure, and in the figure caption. Also would be ideal to display the word "WRF-Chem NO2" directly on the top center of the panel a, and "OMI NO2" on the top center of panel b. And maybe "WRF-Chem - OMI" on panel c.*

We have added the corresponding titles and we have added $NO_2$ to the caption and added the suggested titles to the panels.

*-Figures A1 & C1: It's unclear why these are in an Appendix. If possible, it would be nice to include these in the main text when they are referenced. There is certainly room for them if the journal rules allow it.*

We have replaced the figures to the main text and adjusted the text/references where needed.

[revised manuscript text omitted]